# Simulating *Ips typographus (L.),* outbreak dynamics and their influence on carbon balance estimates with ORCHIDEE r8627

Guillaume Marie[1]*, Jina Jeong[2]*, Hervé Jactel[3], Gunnar Petter[4], Maxime Cailleret[5], Matthew J. McGrath[1], Vladislav Bastrikov[6], Josefine Ghattas[7], Bertrand Guenet[8], Anne Sofie Lansø[9], Kim Naudts[11], Aude Valade[10], Chao Yue[12], Sebastiaan Luyssaert[2]

[1] Laboratoire des Sciences du Climat et de l'Environnement, CEA CNRS UVSQ UP Saclay, 91191 Orme des Merisiers, Gif-sur-Yvette, France

[2] Faculty of Science, A-LIFE, Vrije Universiteit Amsterdam,  1081 BT Amsterdam, the Netherlands

[3] INRAE, University of Bordeaux, UMR Biogeco, 33612 Cestas, France

[4] ETH Zürich, Department of Environmental Systems Science, Forest Ecology, 8092 Zürich, Switzerland

[5] INRAE, Aix-Marseille Univ, UMR RECOVER, 13182 Aix-en-Provence, France

[6] Science Partner, France

[7] Institut Pierre-Simon Laplace – Sciences du climat (IPSL), 75105 Jussieu, France

[8] Laboratoire de Géologie, Ecole Normale Supérieure, CNRS, PSL Research University, IPSL, 75005 Paris, France

[9] Department of Environmental Science, Aarhus Universitet, Frederiksborgvej 399, 4000 Roskilde, Denmark

[10] Eco & Sols, Univ Montpellier, CIRAD, INRAE, 34060 Institut Agro, IRD, Montpellier, France

[11] Department of Earth Sciences, Vrije Universiteit Amsterdam, 1081 HV Amsterdam, the Netherlands

[12] State Key Laboratory of Soil Erosion and Dryland Farming on the Loess Plateau, Northwest A & F University, Yangling, Shaanxi, China

* These authors contributed equally to this study

**Corresponding author:** Guillaume Marie, guillaume.marie@lsce.ipsl.fr, Jina Jeong, j.jeong@vu.nl, Sebastiaan Luyssaert,  s.luyssaert@vu.nl

**Abstract** : New (a)biotic conditions resulting from climate change are expected to change disturbance dynamics, such as windthrow, forest fires, droughts, and insect outbreaks, and their interactions. These unprecedented natural disturbance dynamics might alter the capability of forest ecosystems to buffer atmospheric $CO_2$ increases, potentially leading forests to transform from sinks into sources of $CO_2$. This study aims to enhance the ORCHIDEE land surface model to study the impacts of climate change on the dynamics of the bark beetle *Ips typographus* and subsequent effects on forest functioning. The *Ips typographus* outbreak model is inspired by previous work from Temperli et al. 2013 for the LandClim landscape model. The new implementation of this model in ORCHIDEE r8627 accounts for key differences between ORCHIDEE and LandClim: (1) the coarser spatial resolution of ORCHIDEE, (2) the higher

temporal resolution of ORCHIDEE, and (3) the pre-existing process representation of windthrow, drought, and forest structure in ORCHIDEE. Simulation experiments demonstrated the capability of ORCHIDEE to simulate a variety of post-disturbance forest dynamics observed in empirical studies. Through an array of simulation experiments across various climatic conditions and windthrow intensities, the model was tested for its sensitivity to climate, initial disturbance, and selected parameter values. The results of these tests indicated that with a single set of parameters, ORCHIDEE outputs spanned the range of observed dynamics. Additional tests highlighted the substantial impact of incorporating *Ips typographus* outbreaks on carbon dynamics. Notably, the study revealed that modeling abrupt mortality events, as opposed to a continuous mortality framework, provides new insights into the short-term carbon sequestration potential of forests under disturbance regimes by showing that the continuous mortality framework tends to overestimate the carbon sink capacity of forests in the 20 to 50 year range in ecosystems under high disturbance pressure, compared to scenarios with abrupt mortality events. This model enhancement underscores the critical need to include disturbance dynamics in land surface models to refine predictions of forest carbon dynamics in a changing climate.

## 1. Introduction

Future climate will likely bring new abiotic constraints through the co-occurrence of multiple connected hazards, e.g., "hotter droughts", which are droughts combined with heat waves (Allen et al., 2015; Zscheischler et al., 2018), but also new biotic conditions from interacting natural and anthropogenic disturbances, e.g., insect outbreaks following windthrow or forest fires (Seidl et al., 2017). Unprecedented natural disturbance dynamics might alter biogeochemical cycles specifically the capability of forest ecosystems to buffer the $CO_2$ increase in the atmosphere (Hicke et al., 2012; Seidl et al., 2014) and the risk that forests are transformed from sinks into sources of $CO_2$ (Kurz et al., 2008a). The magnitude of such alteration, however, remains uncertain principally due to the lack of impact studies that include disturbance regime shifts at global scale (Seidl et al., 2011).

Land surface models are used to study the relationships between climate change and the biogeochemical cycles of carbon, water, and nitrogen (Ciais et al., 2005; Cox et al., 2000; Friedlingstein et al., 2006; Luyssaert et al., 2018; Zaehle and Dalmonech, 2011). Many of these models use background mortality to obtain an equilibrium in their biomass pools. This classic approach towards forest dynamics, which assumes steady-state conditions over long periods of time, may not be suitable for assessing the impacts of disturbances on shorter time scales under a fast changing climate. This could be considered a shortcoming in the land surface models because disturbances can have significant impacts on ecosystem services, such as water regulation, carbon sequestration, and biodiversity (Quillet et al., 2010). Mechanistic approaches that account for a variety of mortality causes, such as age, size, competition, climate, and disturbances, are now being considered and tested to simulate forest dynamics more accurately (Migliavacca et al., 2021). For example, the land surface model ORCHIDEE accounts for mortality from interspecific competition for light in addition to background mortality (Naudts et al., 2015). Implementing a more mechanistic view on mortality is thought to be essential for improving our understanding of the impacts of climate change on forest dynamics and the provision of ecosystem services.


Land surface models also face the challenge of better describing mortality particularly when it comes to ecosystem
responses to "cascading disturbances", where legacy effects from one disturbance affect the next (Buma, 2015;
Zscheischler et al., 2018). Biotic disturbances, such as bark beetle outbreaks, strongly depend on previous
disturbances as their infestation capabilities are higher when tree vitality is low, for example following drought or
storm events (Seidl et al., 2018). This illustrates how interactions between biotic and abiotic disturbances can have
substantial effects on ecosystem dynamics and must be accounted for in land surface models to improve our
understanding of the impacts of climate change on forest dynamics (Seidl et al., 2011; Temperli et al., 2013a). While
progress has been made towards including abrupt mortality from individual disturbance types such as wildfire
(Lasslop et al., 2014; Migliavacca et al., 2013; Yue et al., 2014), windthrow (Chen et al., 2018) and drought (Yao et
al., 2022), the interaction of biotic and abiotic disturbances remains both a knowledge and modeling gap (Kautz et
al., 2018).

Bark beetle infestations are increasingly recognized as disturbance events of regional to global importance (Bentz et
al., 2010; Kurz et al., 2008b; Seidl et al., 2018). Notably, a bark beetle outbreak ravaged over 90% of Engelmann
spruce trees across approximately 325,000 hectares in the Canadian and American Rocky Mountains between 2005
and 2017 (Andrus et al., 2020). In Europe, the bark beetle *Ips typographus* has been involved in up to 8% of total
tree mortality due to natural disturbances from 1850 to 2000 (Hlásny et al., 2021a). A recent increase in beetle
activity, particularly following mild winters (Andrus et al., 2020; Kurz et al., 2008c), windthrow (Mezei et al.,
2017), and droughts (Nardi et al., 2023) have been well-documented (Hlásny et al., 2021a; Pasztor et al., 2014),
underscoring the need to integrate bark beetle dynamics into land surface modeling.

Past studies used a variety of approaches to model the impacts of bark beetles on forests. While some models treated
bark beetle outbreaks as background mortality (Luyssaert et al., 2018; Naudts et al., 2016), others dynamically
modeled these outbreaks within ecosystems (Jönsson et al., 2012; Seidl and Rammer, 2016; Temperli et al., 2013b).
Studies with prescribed beetle outbreaks tend to focus on the direct effects of the outbreak on forest conditions and
carbon fluxes, but are likely to overlook more complex feedback processes, such as interactions with other
disturbances and longer-term impacts. Conversely, dynamic modeling of beetle outbreaks, provides a more
comprehensive view by incorporating the lifecycle of bark beetles, tree defense mechanisms, and ensuing alterations
in forest composition and functionality.

Simulation experiments for *Ips typographus* outbreaks using the LPJ-GUESS vegetation model highlighted regional
variations in outbreak frequencies, pinpointing climate change as a key exacerbating factor (Jönsson et al., 2012).
Simulation experiments with the iLand landscape model suggested that almost 65% of *Ips typographus* outbreaks are
aggravated by other environmental drivers (Seidl and Rammer, 2016). A 4°C temperature increase could result in a
265% increase in disturbed area and a 1800% growth in the average patch size of the disturbance (Siedl and Rammer
2016). Disturbance interactions were ten times more sensitive to temperature changes, boosting the disturbance
regime's climate sensitivity. The results of these studies justify the inclusion of interacting disturbances in land
surface models, such as ORCHIDEE, which are used in future climate predictions and impact studies (Boucher et al.,
112 2020).
The objectives of this study are: (1) to develop and implement a spatially implicit outbreak model for *Ips*
*typographus* in the land surface model ORCHIDEE inspired by the work from Temperli et al. (2013), and (2) use
simulation experiments to characterize the behavior of this newly added model functionality.

## 2. Model description

### 2.1. The land surface model ORCHIDEE

ORCHIDEE is the land surface model of the IPSL (Institut Pierre Simon Laplace) Earth system model (Boucher et
al., 2020; Krinner et al., 2005). ORCHIDEE can, however, also be run uncoupled as a stand-alone land surface
model forced by temperature, humidity, pressure, precipitation, and wind fields. Unlike the coupled setup, which
needs to run on the global scale, the stand-alone configuration can cover any area ranging from a single grid point to
the global domain. In this study ORCHIDEE was run as a stand-alone land surface model.
ORCHIDEE does not enforce any particular spatial resolution. The spatial resolution is an implicit user setting that
is determined by the resolution of the climate forcing (or the resolution of the atmospheric model in a coupled
configuration). ORCHIDEE can run on any temporal resolution. This apparent flexibility is somewhat restricted as
processes are formalized at given time steps: half-hourly (e.g., photosynthesis and energy budget), daily (i.e., net
primary production), and annual (i.e. vegetation demographic processes). With the current model architecture
meaningful simulations should have a temporal resolution of one minute to one hour for the calculation of energy
balance, water balance, and photosynthesis.
ORCHIDEE utilizes meta-classes to discretize the  global diversity in vegetation. The model includes 13 meta-
classes by default, including one class for bare soil, eight classes for various combinations of leaf-type and climate
zones of forests, two classes for grasslands, and two classes for croplands. Each meta-class can be further subdivided
into an unlimited number of plant functional types (PFTs). The current default setting of ORCHIDEE distinguishes
15 PFTs where the meta-class of C3 grasslands have been separated into a boreal, temperate and tropical C3
grassland PFT.
At the beginning of a simulation, each forest PFT in ORCHIDEE contains a monospecific forest stand that is
structured by a user-defined but fixed number of diameter classes (three by default). Throughout the simulation, the
boundaries of the diameter classes are adjusted to accommodate changes in the stand structure, while the number of
classes remains constant. Flexible class boundaries provide a computationally efficient approach to simulate
different forest structures. For instance, an even-aged forest is simulated by using a small diameter range between
the smallest and largest trees, resulting in all trees belonging to the same stratum. Conversely, an uneven-aged forest
is simulated by applying a wide range between diameter classes, such that different classes represent different
canopy strata.

The model uses allometric relationships to link tree height and crown diameter to stem diameter. Individual tree
canopies are not explicitly represented, instead a canopy structure model based on simple geometric forms (Haverd
et al. 2012) has been included in ORCHIDEE (Naudts et al., 2015). Diameter classes represent trees with different
mean diameter and height, which informs the user about the social position of trees within the canopy. Intra-stand
competition is based on the basal area of individual trees, which accounts for the fact that trees with a higher basal
area occupy dominant positions in the canopy and are therefore more likely to intercept light and thus contribute
more to stand-level photosynthesis and biomass growth compared to suppressed trees (Deleuze et al., 2004). If
recruitment occurs, diameter classes evolve into cohorts. However, in the absence of recruitment, all diameter
classes contain trees of the same age.

Individual tree mortality from self-thinning, wind storms, and forest management is explicitly simulated. Other
sources of mortality are implicitly accounted for through a so-called constant background mortality rate.
Furthermore, age classes (four by default) can be used after land cover change, forest management, and disturbance
events to explicitly simulate the regrowth of the forest. Following a land cover change, biomass and soil carbon
pools (but not soil water columns) are either merged or split to represent the various outcomes of a land cover
change. The ability of ORCHIDEE to simulate dynamic canopy structures (Chen et al., 2016; Naudts et al., 2015b;
Ryder et al., 2016), a feature essential to simulate both the biogeochemical and biophysical effects of natural and
anthropogenic disturbances, is exploited in other parts of the model, i.e., precipitation interception, transpiration,
energy budget calculations, the radiation scheme, and the calculation of the absorbed light for photosynthesis.

Since revision 7791, mortality from *Ips typographus* outbreaks is explicitly accounted for and thus conceptually
excluded from the so-called environmental background mortality. Subsequently, changes in canopy structure
resulting from growth, forest management, land cover changes, wind storms, and *Ips typographus* outbreaks are
accounted for in the calculations of the carbon, water, and energy exchanges between the land surface and the
atmosphere. For details on the functionality of the ORCHIDEE model that is not of direct relevance for this study,
e.g., energy budget calculations, soil hydrology, snow phenology, albedo, roughness, photosynthesis, respiration,
phenology, carbon and nitrogen allocation, land cover changes, product use, and the nitrogen cycle are readers are
referred to Krinner et al., 2005; Zaehle and Friend, 2010; Naudts et al., 2015; Vuichard et al., 2019.

**2.2.       Origin of the bark beetle (*Ips typographus*) model: the LANDCLIM legacy**
Although mortality from windthrow (Yi-Ying et al., 2018) and forest management (Luyssaert et al., 2018; Naudts et
al., 2016) were already accounted for in ORCHIDEE prior to r8627, insect outbreaks and their interaction with other
disturbances were not. The LandClim model (Schumacher, 2004) and more specifically the *Ips typographus* model
developed by Temperli et al. (2013) has been used as basis to develop the *Ips typographus* model in ORCHIDEE
r8627.

LandClim is a spatially explicit stochastic landscape model in which forest dynamics are simulated at a yearly time
step for 10–100 km² landscapes consisting of 25 m × 25 m patches. Within a patch recruitment, growth, mortality
and competition among age cohorts of different tree species are simulated with a gap model (Bugmann, 1996) in
response to monthly mean temperature, climatic drought, and light availability. LandClim, for which a detailed
description can be found in (Schumacher, 2004; Temperli et al., 2013), includes the functionality to simulate the
decadal dynamics and consequences of *Ips typographus* outbreaks at the landscape-scale (Temperli et al., 2013). In
the LandClim approach, the extent, occurrence and severity of beetle-induced tree mortality are driven by the
landscape susceptibility, beetle pressure, and infested tree biomass. While the LandClim beetle model was designed
and structured to be generally applicable for northern hemisphere climate-sensitive bark beetle-host systems, it was
parameterized to represent disturbances by the *Ips typographus* in Norway spruce (*Picea abies* Karst.; Temperli et
al. 2013).

As LandClim and ORCHIDEE are developed for different purposes, their temporal and spatial scales differ. These
differences in model resolution justify developing a new model while following the principles embedded in the
LandClim approach. LandClim assesses bark beetle damage at 25 m x 25 m patches and to do so it uses information
from other nearby patches as well as landscape characteristics such as slope, aspect and altitude. The susceptibility
of a landscape to bark beetle infestations is calculated using multiple factors such as drought-induced tree resistance,
age of the oldest spruce cohort, proportion of spruce in the patch's basal area, and spruce biomass damaged by
windthrow. These drivers are presented as sigmoidal relationships ranging from 0 to 1 (denoting none to maximum
susceptibility respectively) that are combined in a susceptibility index for each Norway spruce cohort in a patch.
Bark beetle pressure is quantified as the potential number of beetles that can infest a patch, and its calculation
considers, among others, previous beetle activity, maximum possible spruce biomass that beetles could kill, and
temperature-dependent bark beetle phenology. Finally, the susceptibility index and beetle pressure are used to
estimate the total infested tree biomass and total biomass killed by bark beetles for each cohort within a patch.

In ORCHIDEE, however, the simulation unit is about six orders of magnitude larger, i.e. 25 km x 25 km. Hence, a
single gridcell in ORCHIDEE exceeds the size of an entire landscape in LandClim. Where landscape characteristics
in LandClim can be represented by statistical distributions, the same characteristics in ORCHIDEE are represented
by single values. These differences between LandClim and ORCHIDEE imply that the original bark beetle model
cannot be implemented in ORCHIDEE without substantial adjustments;  the model at the ORCHIDEE gridcell
should work without requiring spatial information and statistical distributions of landscape characteristics because
those are not available in ORCHIDEE.



    **2.3.    Bark beetle outbreak development stages**

Bark beetle outbreak development stages (Fig. 1) are useful to understand the dynamics of an outbreak (Edburg et
al., 2012; Hlásny et al., 2021a; Wermelinger, 2004). Nonetheless, the outbreak model in ORCHIDEE r8627
simulates the dynamic of the Ips *typographus* outbreak as a continuous process. Hence, endemic, epidemic, build-up
and post-epidemic stages are not explicitly simulated. In this study, outbreak development stages were only
introduced to structure the model description. If needed, these stages could be distinguished while post-processing
the simulation results if (arbitrary) thresholds are set for specific variables such as $DR_{beetles}$.

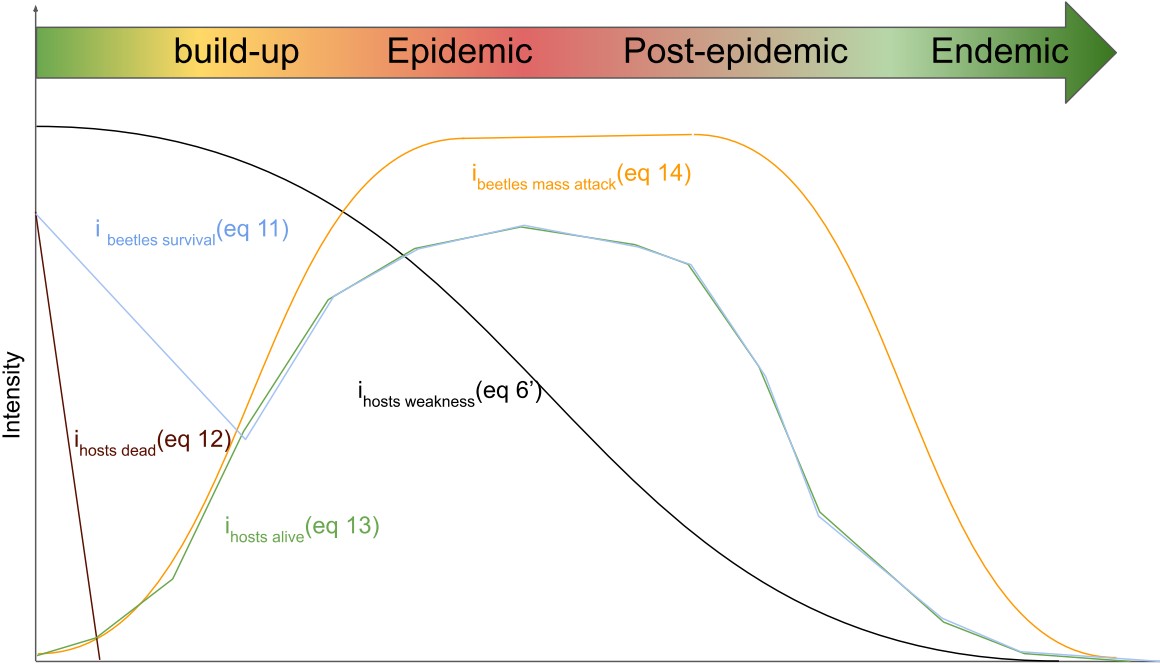

**Figure 1 : Dynamic interplay of the different host and beetle characteristics during a bark beetle (*Ips typographus*) outbreak. The time window spans four outbreak development stages: build-up, epidemic, post-epidemic, and endemic. The curves represent key characteristics, showing the growth in beetle population and subsequent decline in host population. $I_{hosts\ dead}$ characterizes the presence of defenseless uprooted or cut spruce trees; $i_{hosts\ alive}$, characterizes living spruce trees that could become hosts for the bark beetles; $i_{hosts\ susceptibility}$, susceptibility of spruce trees to bark beetle attack; $i_{beetles\ mass\ attack}$, quantifies the capability of the bark beetles to mass attack; $i_{beetles\ survival}$, characterizes the survival of bark beetles. Host and bark beetle characteristics are detailed in the subsequent text. When the density of the host trees is declining due to an increased host mortality from the bark beetle outbreak itself, the competition between trees for light and nutrients declines as well. As a consequence, the host susceptibility decreases which in ORCHIDEE is the main pathway for an outbreak to move back to the endemic phase. After 1 year the wood from a storm is not fresh enough for bark beetles to breed in. In ORCHIDEE, the bark beetle population needs to be capable of mass attacking living trees within a year to make the transition from the build-up to the epidemic phase.**


**2.4.      bark beetle (*Ips typographus*) damage in ORCHIDEE**

**Table 1: List of symbols**

| Symbol | Description | Units |
|---|---|---|
| $\alpha$ | Intercept of the self thinning relationship | unitless |
| $\beta$ | Exponent of the self thinning relationship | unitless |
| $act_{limit}$ | $B_{kill}/B_{total}$ at which $i_{beetles\ activity} = 0.5$ | $gC.m^{-2}$ |
| $B_{beetles\ kill}$ | Biomass of spruce killed by bark beetle annually | $gC.m^{-2}$ |
| $B_{windthrow\ kill}$ | Biomass of spruce killed by windthrow event | $gC.m^{-2}$ |
| $B_{beetles\ attacked}$ | Biomass of spruce attacked by bark beetle annually | $gC.m^{-2}$ |
| $B_{total}$ | Total living biomass of spruce stand | $gC.m^{-2}$ |
| $B_{wood}$ | Woody biomass of spruce stand | $gC.m^{-2}$ |
| $BP_{limit}$ | $i_{beetle\ pressure}$ at which $i_{beetles\ mass\ attack} = 0.5$ | unitless |
| $D_{max}$ | Maximum stand density | $tree.ha^{-1}$ |
| $D_{age\ class}$ | Stand tree density of spruce age classes | $tree.ha^{-1}$ |
| $D_{spruce}$ | Stand tree density of spruce | $tree.ha^{-1}$ |
| $DD_{eff}$ | Cumulative effective degrees days | $°C.Day^{-1}$ |
| $DD_{ref}$ | Reference degrees days to complete one beetle generation | $°C.Day^{-1}$ |
| $Dia_{quadratic}$ | Mean quadratic diameter | meters |
| $DR_{beetles}$ | $B_{beetles\ kill}/B_{total} * 100$ | % |
| $DR_{windthrow}$ | $B_{windtrow\ kill}/B_{total} * 100$ | % |
| $F_{spruce}$ | Area fraction of spruce within gridcell | unitless |
| $F_{age\ class}$ | Area fraction of spruce age classes | unitless |
| $F_{non-spruce}$ | Non-spruce area fraction | unitless |
| $G_{limit}$ | Beetles generation number at which $i_{beetle\ generation} = 0.5$ | Generation |
| $I_{hosts\ competition}$ | Spruce trees under competition pressure | unitless |
| $I_{hosts\ susceptibility}$ | Spruce trees susceptibility to bark beetle attack | unitless |
| $I_{hosts\ attractivity}$ | Spruce attractivity for bark beetles | unitless |
| $I_{hosts\ dead}$ | Defenseless spruce trees uprooted or cut | unitless |
| $I_{hosts\ alive}$ | Potential living hosts for bark beetle | unitless |
| $I_{hosts\ defense}$ | Spruce trees capability to resist a bark beetle attack | unitless |
| $I_{hosts\ share}$ | Spruces hidden by other species to bark beetle detection | unitless |
| $I_{hosts\ competition,\ age\ class}$ | Spruce age class under competition pressure | unitless |
| $I_{hosts\ defense,\ age\ class}$ | Spruce age class capability to resist a bark beetle attack | unitless |
| $I_{hosts\ health,\ age\_class}$ | Spruce age class health condition | unitless |
| $I_{beetles\ pressure}$ | Proxy of bark beetle population level | unitless |
| $I_{beetles\ survival}$ | Bark beetle survival index | unitless |
| $I_{beetles\ generation}$ | Bark beetle generation index | unitless |
| $i_{beetles\_activity}$ | Previous bark beetles activity index | unitless |
| $i_{beetles\_mass\_attack}$ | Bark beetles mass attack capability | unitless |
| $max_{Nwood}$ | Value of $N_{wood}$ at which $i_{hosts\ dead} = 1.0$ | unitless |
| $N_{wood}$ | Spruce woody necromass | $gC.m^{-2}$ |
| $P_{success,\ age\_class}$ | Probability of successful attack per age class | unitless |
| $P_{attack}$ | Probability of beetles attack | unitless |
| $PWS_{max}$ | Maximum long term spruce water stress | unitless |
| $PWS_{spruce}$ | Spruce water stress | unitless |
| $PWS_{age\_class}$ | Spruce age classes water stress | unitless |
| $PWS_{limit}$ | Spruce water stress at which $i_{hosts\ defense} = 0.5$ | unitless |
| $i_{rd\_limit}$ | Relative density index at which $i_{hosts\ competition} = 0.5$ | unitless |
| $i_{rd\_susceptibility}$ | Relative density index at which $i_{host\ susceptibility} = 0.5$ | unitless |
| $i_{rd\_spruce}$ | Spruce stand relative density index [0,1] | unitless |
| $I_{rd,\ age\ class}$ | Spruce age classes relative density index [0,1] | unitless |
| $S_{competition}$ | Shape parameter in the calculation of $i_{hosts\ competition}$ | unitless |
| $S_{susceptibility}$ | Shape parameter in the calculation of $i_{hosts\ susceptibility}$ | unitless |

| | | |
|---|---|---|
| $S_{drought}$ | Shape parameter in the calculation of $i_{hosts\ defense}$ | unitless |
| $S_{share}$ | Shape parameter in the calculation of $i_{hosts\ share}$ | unitless |
| $S_{activity}$ | Shape parameter in the calculation of $i_{beetle\ activity,\ y-1}$ | unitless |
| $S_{generation}$ | Shape parameter in the calculation of $i_{beetle\ generation}$ | unitless |
| $Sh_{spruce}$ | Share fraction of spruce against non-spruce in gridcell | unitless |
| $Sh_{limit}$ | Share fraction at which $i_{hosts\ share} = 0.5$ | unitless |
| $T_{air}$ | Air temperature | °C |
| $T_{max}$ | Temperature above which beetles developpement stop | °C |
| $T_{min}$ | Temperature below which beetles developpement stop | °C |
| $T_{bark}$ | Bark temperature | °C |
| $T_{opt}$ | Optimal bark temperature for beetles development | °C |


The biomass of trees killed by bark beetles in one year and one gridcell ($B_{beetles\ kill}$) is calculated as the product of the
biomass of trees attacked by bark beetles ($B_{beetles\ attacked}$) and the probability of a successful attack ($P_{success,\ age\ class}$)
averaged over the number of spruce age classes and weighted by their actual fraction ($F_{age\ class} / F_{spruce}$). The approach
assumes that a successful beetle colonization always results in the death of the attacked tree which is a simplification
from reality (A. Leufvén et al. 1986).

$$B_{beetles\ kill} = \sum_{nb\ age\ classes}^{age\ class=1} P_{success,\ age\ class} \times B_{beetles\ attacked} \times \frac{F_{age\ class}}{F_{spruce}} \qquad (1)$$

During the endemic stage, $B_{beetles\ attacked}$ and $B_{beetles\ kill}$ are at their lowest values and the damage from bark beetles has
little impact on the structure and function of the forest. Losses from $B_{beetles\ kill}$ can be considered to contribute to the
background mortality.

The biomass of trees attacked by bark beetles ($B_{beetles\ attacked}$) is the outcome of bark beetles that successfully overcame
the tree defenses and succeeded in boring holes in the bark in order to reach the sapwood. $B_{beetles\ attacked}$ is calculated at
the gridcell by multiplying the actual stand biomass of spruce ($B_{total}$) and the probability that bark beetles attack
spruce trees in the gridcell ($P_{attacked}$).

$$B_{beetles\ attacked} = B_{total} \times P_{attacked} \qquad (2)$$

$P_{attacked}$ represent the ability of the bark beetles to spread and to locate new suitable spruce trees as hosts for breeding.
$P_{attacked}$ is calculated by the product of two indexes (all indexes in this study are denoted i and are analogue the
susceptibility indexes from Temperli et al. 2013): (1) the beetle pressure index ($i_{beetles\ pressure}$) which a proxy of the bark
beetle population and (2) the stand attractivity index ($i_{hosts\ attractivity}$) is related to its health and reflects the ability of the
forest to resist an external stressor such as bark beetle attacks.

$$P_{attacked} = i_{hosts\ attractivity} \times i_{beetles\ pressure} \qquad (3)$$

## 2.5. Host attractivity

The stand attractivity index ($i_{hosts\ attractivity}$) represents how interesting a stand is for a new bark beetle colony. When $i_{hosts\ attractivity}$ tends to 0, the stand is constituted mainly by healthy trees which are less attractive for beetles whereas an $i_{hosts\ attractivity}$ approaching 1 represents a highly stressed spruce stand suitable for colonization by bark beetles. Factors that contribute to the stress of a forest are: nitrogen limitation, limited carbohydrate reserves, and monospecific spruce forest. Trees experiencing extended periods of environmental stress are expected to have less carbon and nitrogen reserves available for defense compounds, making them vulnerable for bark beetle attacks even at relatively low beetle population densities (Raffa et al., 2008). Nonetheless, reserves pools in ORCHIDEE r8627 have not yet been evaluated so, instead proxies were used such as long term drought ($PWS_{max}$) and relative density index ($i_{rd}$) which were already simulated in ORCHIDEE r8627.

$$i_{hosts\ attractivity} = max\left(i_{hosts\ competition}, i_{hosts\ defense}\right) \times i_{hosts\ share} \qquad (4)$$

Where $i_{hosts\ competition}$ and $i_{hosts\ defense}$ both represent proxies for the reduction of the nitrogen and carbohydrate reserve due to strong competition for light and soil resources, and consecutive years that are drier than average. For this study, the max drought intensity during the last three years ($PWS_{max}$) is considered, as a proxy of spruce stand healthiness:

$$i_{hosts\ defense} = 1/\left(1+e^{S_{drought}\cdot\left(1-PWS_{max}-PWS_{limit}\right)}\right) \qquad (5a)$$

$$PWS_{max} = \sum_{nb\ age\ class=3}^{age\ class=1} max\left(PWS_{spruce,n}, PWS_{spruce,n-1}, PWS_{spruce,n-2}\right) \times \frac{F_{spruce\ class}}{F_{spruce}} \qquad (5b)$$

Where $PWS_{spruce}$ is the average daily plant water stress index over the growing season for the spruce stand and is equal to 0 when plants are highly stressed. $PWS_{limit}$ is the plant water stress below which the healthiness of the stand will strongly be affected. *Nb age class* is the numbers age class within the stand and is equal to 3 in this study. In addition to drought, overstocked forest may also decrease the overall healthiness of a spruce stand ($i_{hosts\ competition}$).

$$i_{hosts\ competition} = 1/\left(1+e^{S_{competition}\cdot\left(i_{rd\ spruce}-i_{rd\ limit}\right)}\right) \qquad (6a)$$

In ORCHIDEE, the relative density index ($i_{rd}$) is used to quantify the competition between trees at the stand level. At an $i_{rd}$ of 1, the forest is expected to be at its maximum density given the carrying capacity of the site, implying the highest level of competition between trees. $i_{rd\ limit}$ represents the limit at which the bark beetle outbreak starts to decline because of lack of suitable host trees. The severity of bark beetle-caused tree mortality decreases when we increase the spatial resolution from the stand to the landscape scale. At the landscape scale, which can cover areas up to 2500 km², the duration of mortality may be longer and the severity lower because beetles disperse across the landscape and cause mortality at different times. This distinction is important for interpreting model results,

particularly when considering parameters like $i_{rd\ limit}$ in the ORCHIDEE model. $i_{rd\ limit}$ describes the proportion of trees
surviving after an outbreak and should therefore be adjusted for the spatial scale of a gridcell in ORCHIDEE. In
model set-up where a gridcell represents a single stand (~1 ha), $i_{rd\ limit}$ should be close to 0, indicating that nearly all
trees may be killed. However, in a simulation with gridcells representing 2500 km², not all trees will be killed, which
is reflected in setting $i_{rd\ limit}$ to 0.4.

$i_{rd\ spruce}$ is computed as follows:

$$i_{rd\ spruce} = \sum_{nb\ age\ class=3}^{age\ class=1} \frac{D_{age\ class}}{D_{max}} \times \frac{F_{age\ class}}{F_{spruce}} \tag{6b}$$

Where $D_{age\ class}$ is the current tree density of an age class and $F_{age\ class}$ is the fraction of spruce in the gridcell that
resides in this age class. $D_{max}$ represents the maximum stand density of a stand given its diameter. In ORCHIDEE
$D_{max}$ is calculated based on the quadratic mean diameter (cm) of the age class and two species specific parameters, $\alpha$
and $\beta$:

$$D_{max} = \left(Di\,a_{quadratic,age\ class}/\alpha\right)^{(1/\beta)} \tag{6c}$$

The index $i_{hosts\ share}$ (used in eq. 4) takes into account that in a mixed tree species landscape, even a few non-host trees
may chemically hinder bark beetles in finding their host trees (Zhang and Schlyter, 2004) explaining why insect
pests, including *Ips typographus* outbreaks, often cause more damage in pure compared to mixed stands (Nardi et al.,
2023). ORCHIDEE r8627 does not simulate multi-species stands but does account for landscape-level heterogeneity
of forests with different plant functional types. The bark beetle model in ORCHIDEE assumes that within a gridcell,
the fraction of spruce over other tree species is a proxy for the degree of mixture:

$$i_{hosts\ share} = 1/\left(1+e^{S_{share}\cdot(SH_{spruce}-SH_{limit})}\right) \tag{7a}$$

Where,

$$Sh_{spruce} = F_{non-spruce}/F_{spruce} \tag{7b}$$

### 2.6. Implicit representation of bark beetle populations

The bark beetle pressure Index $(i_{beetles\ pressure})$ is now formulated based on two components: (1) the bark beetle
breeding index of the current year $(i_{beetles\ generation})$, and (2) an index of the loss of tree biomass in the previous year due
to bark beetle infestation $(i_{beetles\ activity})$. $i_{beetles\ activity}$ is thus a proxy of the previous year's bark beetle activity. The
expression accounts for the legacy effect of bark beetle activities by averaging activities over the current and
previous years. In this approach, the susceptibility index ($i_{beetles\ survival}$) serves as an indicator for increased bark beetle
survival which could result from favorable conditions for beetle demography (see next section).

$i_{beetles\ pressure} = i_{beetles\ survival} \times \dfrac{(i_{beetles\ generation} + i_{beetles\ activity})}{2}$ \hfill (8)

The model calculates $i_{beetles\ generation}$ from a logistic function, which depends on the number of generations a bark beetle
population can sustain within a single year:

$i_{beetles\ generation} = 1/\left(1 + e^{-S_{generation} \cdot \left(\frac{DD_{eff}}{DD_{ref}} - G_{limit}\right)}\right)$ \hfill (9)

Where $S_{generation}$ and $G_{limit}$ are tuning parameters for the logistic function, $DD_{eff}$ represents the sum of effective
temperature for bark beetle reproduction in °C.Day$^{-1}$, while $DD_{ref}$ denotes the thermal sum of degree days for one
bark beetle generation in °C.Day$^{-1}$. Saturation of $i_{beetles\ generation}$ represents the lack of available breeding substrate when
many generations develop over a short period.

$DD_{eff}$ is calculated from January 1$^{st}$ until the diapause of the first generation. In ORCHIDEE, diapause is triggered
when daylength exceeds 14.5 hours (e.g., April 27$^{th}$ for France). Each day before the diapause with a daily average
temperature around the bark above 8.3°C ($T_{min}$) and below 38.4°C ($T_{max}$) is accounted for in the summation of $DD_{eff}$
(eq.10). This approach simulates the phenology of bark beetles, which tend to breed earlier when winter and spring
were warmer, thus allowing for multiple generations in the same year (Hlásny et al., 2021a).
$DD_{eff} = \displaystyle\sum_{n_{diapause}}^{i=1} (T_{opt} - T_{min}) \cdot e^{(0.0288 \cdot T_{bark,i})} - e^{(0.0288 \cdot b_{eff} - (40.99 - T_{bark,i})/3.59)} - 1.25$ (10)

Where $i$ is a day, $n_{diapause}$ is the number of days between the 1st of january and the day of the diapause. $T_{opt}$ (30.3°C) is
the optimal bark temperature for beetles development and $T_{min}$ (8.3°C) is the temperature below which the beetles
developpement stop. $T_{bark,\ i}$ is the average daily bark temperature. $T_{bark,\ i}$ is calculated as the daily average air
temperature minus 2°C. All parameters values are taken from Temperli et al. 2013
The bark beetle activity of the previous year ($i_{beetles\ activity}$) is calculated as:

$i_{beetles\ activity} = 1/\left(1 + e^{-S_{activity}\left(\frac{B_{kill,\ y-1}}{B_{total}} - act_{limit}\right)}\right)$ \hfill (11)

Where $i_{beetles\ activity}$ denotes the biomass of the stand damaged by bark beetles in the previous year, $B_{total}$ is the total
biomass of the stand, and $S_{activity}$ and $act_{limit}$ are parameters that drive the intensity of this negative feedback.

During the build-up stage the population of bark beetles can either return to its endemic stage if tree defense
mechanisms are preventing bark beetles from successfully attacking healthy trees, or evolve into an epidemic stage
(Fig. 1) if the tree defense mechanisms fail. During the post-epidemic stage, the forest is still subject to higher
mortality than usual but signs of recovery appear (Hlásny et al., 2021a). Recovery may help the forest ecosystem to
return to its original state or switch to a new state (different species, change in the forest structure) depending on the
intensity and the frequency of the disturbance (Van Meerbeek et al., 2021).
**2.7.    Bark beetle survival**
The capability of the bark beetles to survive the winter in between two breeding seasons is critical in simulating
epidemic outbreaks. During regular winters, winter mortality for bark beetles is around 40% for the adults and 100%
for the juveniles (Jönsson et al. 2012). In our scheme, this mortality rate is implicitly accounted for in the calculation
of the bark beetle survival index ($i_{beetles\ survival}$). A lack of data linking bark beetle survival to anomalous winter
temperatures, justifies the implicit approach and prevented including this information as a modulator of $i_{beetles\ survival}$.
The latter explains why winter temperatures do not appear in eq. 11. Instead the model simulates the survival as a
function of the abundance of suitable tree hosts which decreases the competition for shelter and food:
$i_{beetles\ survival} = max\left(i_{hosts\ dead}, i_{hosts\ alive}\right)$               (12)
The availability of wood necromass from trees that died recently, particularly following windstorms, plays a critical
role in bark beetle survival and proliferation. In the year following a windstorm, uprooted and broken trees may offer
an ideal breeding substrate for bark beetles, facilitating their population growth.
In Temperli et al. (2013) an empirical correlation between windthrow events and bark beetle susceptibility was
established. ORCHIDEE enhances realism by considering the actual suitable hosts (living or recently dead trees) as
the primary driver of bark beetle survival. To avoid overestimating bark beetle population growth, $max_{Nwood}$ has been
introduced. Any addition of dead trees beyond $max_{Nwood}$ is considered ineffective in affecting the bark beetle
population. This ensures that an excess of breeding substrate does not artificially inflate beetle numbers.
This relationship is quantitatively represented in ORCHIDEE through the dead host index, $i_{hosts\ dead}$, which is driven
by the availability of recent dead trees. The formulation of $i_{hosts\ dead}$ is as follows:
$i_{hosts\ dead} = min\left(\dfrac{N_{wood}}{B_{wood}}/max_{Nwood}, 1\right)$               (13)
Here, $N_{wood}$ represents the quantity of woody necromass from the current year, $B_{wood}$ is the total living woody biomass
in the stand, and $max_{Nwood}$ is the threshold of the ratio $N_{wood}/B_{wood}$ signifying the maximum level. This index captures
the immediate increase in dead trees suitable for bark beetle breeding following a windthrow event. However, it
takes about a year for dead wood to lose its freshness and suitability for bark beetle breeding. This is accounted for
by excluding woody necromass that is older than 1 year from the $i_{hosts\ dead}$ calculation.

$max_{Nwood}$ can also be considered as a parameter that depends on the spatial scale of the simulation. The mortality rate
of trees ($DR_{windtrow}$) that will trigger an outbreak is very different across spatial scales. Where a relatively high share
of dead wood is needed to trigger an outbreak at the patch-scale, a much lower share of dead wood suffices at the
landscape-scale to trigger a widespread bark beetle outbreak. So these parameters must be set according to the
spatial resolution of the simulation experiment.

$i_{hosts\ alive}$ denotes the survival of bark beetles which is facilitated by the abundance of suitable trees which reduces the
competition among bark beetles for breeding substrates and therefore increases their survival.

$$i_{hosts\ alive} = i_{beetles\ mass\ attack} \times i_{hosts\ susceptibility} \qquad (14)$$

The amount of suitable tree hosts $i_{hosts\ alive}$ is driven by two factors: (1) the abundance of weak trees which can be
more easily infected by bark beetles. ORCHIDEE does not explicitly represent weak trees, but tree health is thought
to decrease with an increasing density given the stand diameter. The index for host suitability is thus calculated by
making use of the relative density index ($i_{rd\ spruce}$).

$$i_{hosts\ susceptibility} = 1/\left(1 + e^{S_{susceptibility} \cdot \left(i_{rd\ spruce} - i_{rd\ susceptibility}\right)}\right) \qquad (6a')$$

Equation 6a' is close to equation 6a but the parameter $S_{susceptibility}$ has been reduced by a factor of two in order to
reflect that $i_{hosts\ susceptibility}$ is more sensitive to $i_{rd\ spruce}$ than $i_{hosts\ competition}$. (2) $i_{hosts\ mass\ attack}$ which represent the ability of bark
beetles to attack healthy trees when the number of bark beetles is large enough. This index only depends on the size
of the bark beetle population ($i_{beetles\ pressure}$ see eq. 8)

$$i_{hosts\ mass\ attack} = 1/\left(1 + e^{S_{mass\ attack} \cdot \left(i_{beetles\ pressure} - BP_{limit}\right)}\right) \qquad (15)$$

Where $S_{hosts\ mass\ attack}$ and $BP_{limit}$ are parameters. $S_{mass\ attack}$ controls the steepness of the relationship while $BP_{limit}$ is the
bark beetle pressure index at which the population is moving from endemic to epidemic stage where mass attacks are
possible.

The epidemic stage corresponds to the capability of bark beetles to mass attack healthy trees and overrule tree
defenses (Biedermann et al., 2019). At this point in the outbreak, all trees are potential targets irrespective of their
health. Three causes have been suggested to explain the end of the epidemic phase: (1) the most likely cause is a
high interspecific competition among beetles for tree host when the density is decreasing (decreasing $i_{hosts\ alive}$)
(Komonen et al., 2011; Pineau et al., 2017), (2) a series of very cold years will decrease their ability to reproduce
(decreasing $i_{beetles\ generation}$), and (3) a rarely demonstrated increasing population of beetle predators (Berryman, 2002).
In ORCHIDEE r8627, the first two causes are represented but the last, i.e. the predators are not.

**2.8.**      **Tree mortality from bark beetle infestation**

When bark beetles attack a tree, the success of their attack will likely depend on the capability of the tree to defend
itself from the attack. Trees defend themselves against beetle attacks by producing secondary metabolites (Huang et
al., 2020). The high carbon and nitrogen costs of these compounds limit their production to periods with
environmental conditions favorable for growth (Lieutier, 2002). The probability of a successful bark beetle attack is
driven by the size of the bark beetle population ($i_{beetles\ pressure}$) and the health of each tree. ORCHIDEE, however, is not
simulating individual trees but rather diameter classes within an age class. An index of tree health for each age class
($i_{hosts\ health,\ age\ class}$) was calculated as:

$$P_{success,\,age\,class} = i_{hosts\,health,\,age\,class} \times i_{beetles\,pressure} \qquad (16)$$

A tree rarely dies solely from bark beetle damage (except during mass attacks) as female beetles often carry blue-
stain fungi, which colonizes the phloem and sapwood, blocking the water-conducting vessels of the tree (Ballard et
al., 1982). This results in tree death from carbon starvation or desiccation. As ORCHIDEE r8627 does not simulate
the effects of changes in sapwood conductivity on photosynthesis and the resultant probability of tree mortality, the
index of weakened trees index ($i_{hosts\ health,\ age\ class}$) makes use of two proxies similarly to equation 5 and 6 but simplified
to be calculated only for one age class at a time:

$$i_{hosts\,health,\,age\,class} = \frac{\left( i_{hosts\,competition,\,age\,class} + i_{hosts\,defense,\,age\,class} \right)}{2} \qquad (17)$$

$$i_{hosts\,defense,\,age\,class} = 1 / \left( 1 + e^{S_{drought} \cdot \left( 1 - PW\,S_{age\,class} - PW\,S_{limit} \right)} \right) \qquad (5a')$$

Contrary to equation 5a, $PWS_{age\ class}$ is the plant water stress from the current year.

$$i_{hosts\,competition,\,age\,class} = 1 / \left( 1 + e^{S_{competition} \cdot \left( i_{rd\,age\,class} - i_{rd\,limit} \right)} \right) \qquad (6a'')$$

$$i_{rd\,age\,class} = \frac{D_{age\,class}}{D_{max}} \qquad (6b'')$$

To access the bark beetle damage rate ($DR_{bettles}$), $B_{beetles\ kill}$ has to be divided by $B_{total}$.


**2.9.    Flow of the calculations**
The equations presented above contain feedback loops which have been visualized in Fig. 2. In ORCHIDEE these
feedback loops are accounted for in subsequent time steps rather than the same time step.

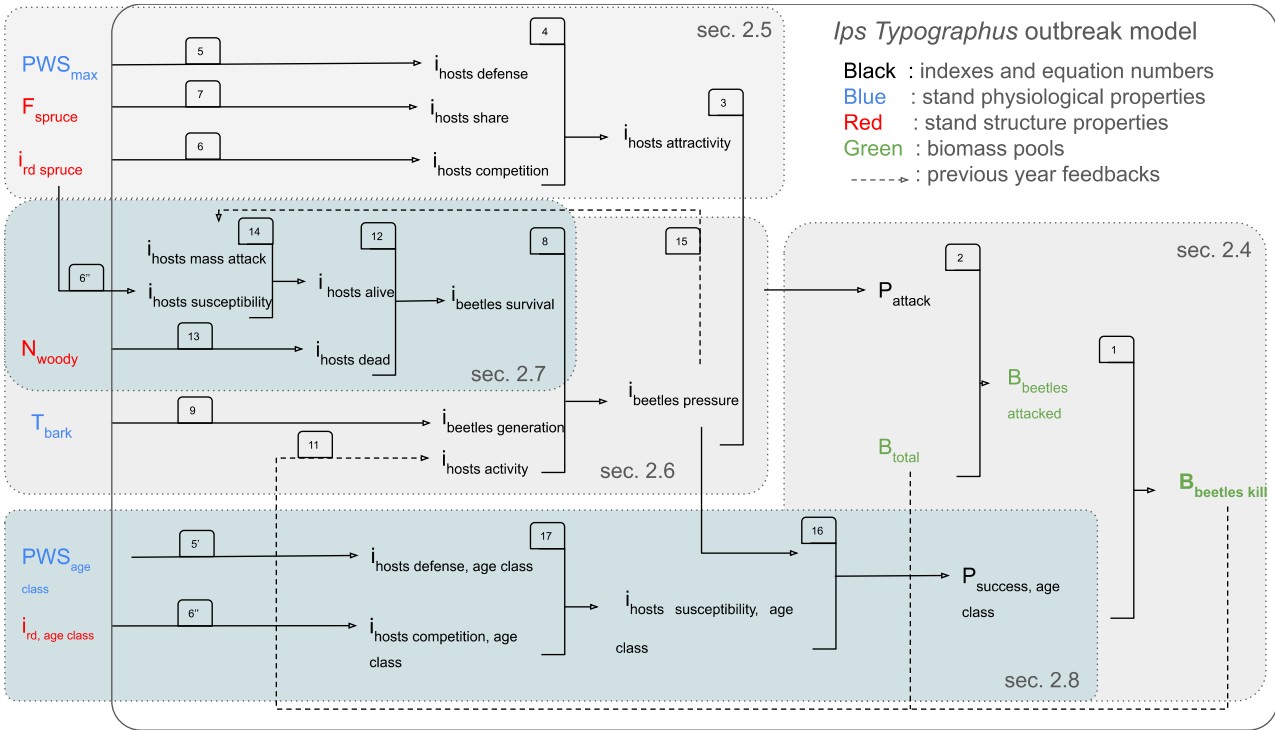

**Figure 2: Order of the calculations and feedback in the *Ips typographus* outbreak model of ORCHIDEE. The numbers correspond to the equation numbers in this study. The dotted line boxes represent 5 main concepts of the outbreak model described in section 2.4, 2.5, 2.6, 2.7, 2.8.**


**3.    Methods and material**
**3.1.    Model configuration**
Given the large-scale nature of the ORCHIDEE, a sensitivity experiment of the bark beetle outbreak functionality
was carried out rather than focusing the model evaluation on matching observed damage volumes at specific case
studies. Focussing on model sensitivity for a range of environmental conditions is thought to reduce the risk of
overfitting the model to specific site conditions (Abramowitz et al., 2008).

ORCHIDEE r8627 including the bark beetle model was run at the location of eight FLUXNET sites, selected to
simulate a credible temperature and precipitation gradient for spruce (see below). For each location, the half-hourly
meteorological data from the flux tower were gap filled and reformatted so that they could be used as climate forcing
by the ORCHIDEE. Boundary conditions for ORCHIDEE, such as soil texture, pH and soil color were retrieved
from the USDA map, for the corresponding gridcell. The observed land cover and land use for the gridcell were
ignored and set to pure spruce because this study did not investigate the effect of species mixture in the simulation
experiments. The resolution of the gridcell chosen for this analysis is 2500 km². Although this corresponds to a high
resolution for large-scale simulations with ORCHIDEE it is  a coarse resolution for studying bark beetle outbreaks.

The climate forcings were looped over as much as needed to bring the carbon, nitrogen, and water pools to
equilibrium during a 340 years long spinup. Following the spinup, a 100-years simulation was run starting with a
windthrow event on the first day of the first year. The results presented in this study come from the 100-years long
simulations. Given the focus on even-aged monospecific spruce forests in regions where spruce growth is not
constrained by precipitation, variables such as $i_{hosts\ share}$ and $i_{hosts\ defense}$ were omitted from this study. Note that
ORCHIDEE does not account for possible acclimation e.g., temporal changes  in bark beetle behavior or  bark beetle
resistance to external stressor such as winter temperature.

**3.2.    Selection of locations**

Bark beetle populations are known to be sensitive to temperature as they are more likely to survive a mild winter
(Lombardero et al., 2000) and tend to breed earlier when winter and spring are warmer than usual, allowing for
multiple generations in the same year (Hlásny et al., 2021a, also see eq. 10 from section 2.6). In order to assess the
temperature effect of the bark beetle outbreak model in ORCHIDEE, eight locations in Europe were selected (Table
2) which represent the range of climatic conditions within the distribution area of Norway spruce (*Picea Abies* Karst
L.), the main host plant for *Ips typographus,* the bark beetle species under investigation.

**Table 2: Climate characteristics of the eight locations used in the simulation experiments. The acronyms refer to the site names used in the FLUXNET database (Pastorello et al. 2020).**

| Site | FI-HYY | DK-SOR | DE-THA | CZ-WET | FR-HES | FR-FON | IT-REN | IT-COL |
|---|---|---|---|---|---|---|---|---|
| Full name | Hyytiala | Soroe | Tharandt | Třeboň | Hesse | Fontainebleau | Renon | Collelongo |
| Country | Finland | Denmark | Germany | Czech | France | France | Italy | Italy |
| Latitude (°N) | 61.84 | 55.49 | 50.96 | 49.02 | 48.4 | 48.48 | 46.59 | 41.85 |
| Longitude (°E) | 24.29 | 11.64 | 13.57 | 14.77 | 7.1 | 2.78 | 11.43 | 13.59 |
| MAT (°C) | 3.8 | 8.2 | 8.2 | 7.7 | 9.5 | 10.2 | 4.7 | 6.3 |
| MinAT (°C) | -10.8 | 2.7 | -3.9 | -5.2 | 0.1 | -1.1 | -6.3 | -3.8 |
| MAP (mm.y$^{-1}$) | 522 | 811 | 734 | 587 | 653 | 989 | 752 | 1050 |
| Mean annual net radiation (w.m$^{-2}$) | 42.1 | 49.4 | 52.5 | 68.0 | 53.7 | 50.3 | 67.7 | 68.3 |

For these eight locations, half-hourly weather data from the FLUXNET database (Pastorello et al., 2020) were used
to drive ORCHIDEE. Some of these locations (FON, SOR, HES, COL, WET) are in reality not covered by spruce
but all sites are, however, located within the distribution of Norway spruce. In this study, locations were selected to
use the observed weather data to simulate a credible temperature and rainfall gradient for spruce. HES location is no
longer part of the FLUXNET network but the previous data are still available are relevant for this analysis.

**3.3.    Sensitivity to model parameters**

The sensitivity assessment evaluates the responsiveness of four key variables ($i_{hosts\ susceptibility}$, $i_{beetles\ mass\ attack}$, $i_{beetles\ generation}$,
$i_{beetles\ activity}$) of the *Ips typographus* outbreak model implemented in ORCHIDEE. The assessment aims to demonstrate
the ability of ORCHIDEE to simulate diverse dynamics of bark beetle infestations. The selection of $i_{hosts\ susceptibility}$,
$i_{beetles\ activity}$, $i_{beetles\ mass\ attack}$, and $i_{beetles\ generation}$ was based on two criteria: (1) their substantial influence on the dynamics of
the *Ips typographus* outbreak noted during model development, and (2) their independence from direct measurable
data, rendering them less suitable for evaluation through literature review.

For each of the four variables, three distinct values were assigned to two parameters labeled "Shape" and "*Limit*".
The *Shape* parameter determines the shape of the logistic relationship, with three values tested: (a) *Shape=-1.0,*
yielding a linear relationship, (b) *-5.0<Shape<-30.0,* resulting in a logistic curve, and (c) *Shape=-500.0,* turning the
logistic relationship into a step function. For the logistic curve, the exact *Shape* value between -30.0 and -5.0 is
chosen according to each index under study:    (1) $S_{susceptibility}$ = -5.0; (2) $S_{activity}$= -20.0; (3) $S_{mass\ attack}$= -30.0; and (4)
$S_{generation}$=5.0 . For $S_{mass\ attack}$ and $S_{activity}$, higher values have been chosen because the slope of the logistic curve has a
significant impact in order to trigger an outbreak.

The second parameter called "*Limit*" determines the threshold, derived from expert insights, at which the logistic
relationship will reach its midpoint value of 0.5 ($i_{rd\ susceptibility}$, $BP_{limit}$, $Act_{limit}$, or $G_{limit}$). For instance, $i_{rd\ susceptibility}$ is set at
0.55, indicating $i_{hosts\ susceptibility}$ midpoint sensitivity (Eq. 6'). Setting $BP_{limit}$ at 0.12 results in an $i_{beetles\ mass\ attack}$ midpoint
when $i_{beetles\ pressure}$ is 0.12, selected for its proximity to scenarios where $i_{hosts\ dead}$ equals 1.0 (Eq. 14). $Act_{limit}$, was
positioned at 0.06, signifying the $i_{beetles\ activity}$ midpoint at a $DR_{beetles}$ = 6% from the preceding year, exceeding endemic
levels yet not reaching epidemic outbreaks (Eq. 10). Lastly, $G_{limit}$ is fixed at 1.0, denoting the midpoint for $i_{beetles}$
$_{generation}$ upon completing one generation annually, underpinning the rarity of bark beetle outbreaks with fewer than
one generation per year (Eq. 9). Starting from these reference values, a "*restrictive*" simulation was run in which the
"*Limit*" parameter values were reduced by 50%. Likewise a "*permissive*" simulation was run to test 50% higher
values for "*Limit*".

The sensitivity analysis of the model parameters explores 36 (3 shapes x 3 limits x 4 equations) combinations of
parameters values named "set", but the full design of the experiment is $8^3$=512 sets (8 parameters, 3 values for each).
This deliberate choice has been made because of the computation time cost of a single run. In order to reduce the
number of runs from 512 to 36, we had to make simplifications: (1) one equation at the time is studied, reducing to 9
the number of sets necessary to realize the sensitivity analysis (2) every other parameters from the remaining
equation is set to default value e.g. "Limits" are set to their reference values and "shape" are set to their a priori
assumption (table 4). The major drawback of this approach is that interaction effects between equations can not be
investigated in the study. Nonetheless, this sensitivity analysis aims to document model behavior, rather than seeking
precise parameter values which can be achieved with the main effect of each equation only (see section 3.4).
The simulations were run for the THA site, where they were repeated for two prescribed windthrow events with a
different intensity, i.e., a $DR_{windthrow}$ of 0.1 and 10%. The effect of the parameters with a negligible windthrow event,
i.e., killing only 0.1% of the trees, was tested to confirm that the selected parameters did make ORCHIDEE simulate
a bark beetle outbreak in the absence of windthrow (*score5* in section 3.4).

### 3.4. Parameter tuning and credibility score

The results of the sensitivity experiment were used to select key model parameters. Selecting the values for the *Shape* and *Limit* parameters (see section 3.3) used in the calculation of the variables $i_{hosts\ susceptibility}$, $i_{beetles\ mass\ attack}$, $i_{beetles\ generation}$, and $i_{beetles\ activity}$ has been carried out in order to reproduce the observed dynamics of bark beetle outbreaks. Observed dynamics were compiled through a literature search for peer-reviewed papers that reported quantitative characteristics of bark beetle outbreaks (Table 3). Four characteristics could be documented and use to calculate score:

- The delay between the windthrow event and the start of the bark beetle outbreak (*score1*).
- The length of the bark beetle outbreak is defined by the number of years required for a bark beetle population to go back to its endemic level (*score2*).
- The cumulative number of trees per unit area, killed by the bark beetles at the end of an outbreak (*score3*).
- The average tree mortality rate ($DR_{beetles}$) during an endemic stage (*score4*).

Based on Table S1 and the reference range in Table 3, scores are calculated for each parameter set. The Credibility Score (*CS*) is the sum of four scores, indicating that the result falls within the four reference ranges described above and no outbreak is triggered when DRwindthrow = 0.1%. The *CS* is computed as follows: *CS = (score1 + score2 + score3 + score4) x score*5. Only parameter sets achieving a *CS* of 4 will be selected. If multiple parameter values are possible for a given equation, the most frequently selected value will be preferred.

**Table 3 : Literature-based summary of characteristics of large-scale bark beetle outbreaks. Due to data spacity, the characteristics combine outbreak dynamics of different bark beetle species, different host species, and different locations. The reference range is used to calculate the credibility score (CS) of each set of parameters (but see table s1).**

| Outbreak characteristics | Literature findings | Reference range | How to estimate in ORCHIDEE ? |
|---|---|---|---|
| Delay before the start of an outbreak (build-up) | A notable surge in the population of *I. typographus* was observed in windthrow areas during the second to third summer following the storm (Havašová et al., 2017; Kärvemo and Schroeder, 2010; Wermelinger, 2004; Wichmann and Ravn, 2001). | [2, 3] years, use in the calculation of *score1* | Using the tree mortality rate by bark beetles ($DR_{beetles}$), one can access the number of years since the storm before reaching the maximum mortality rate (epidemic stage). |
| Length of an outbreak (epidemic) | Studies suggest that *I. typographus* outbreaks in Europe can last anywhere from 11 to 17 years (Bakke, 1989; Hlásny et al., 2021b; Mezei et al., 2014). | [11, 17] years, use in the calculation of *score2* | Using the tree mortality rate by bark beetles ($DR_{beetles}$), one can access the number of years past since the storm before reaching the minimum mortality rate (endemic stage). |
| Severity rate of an outbreak (severity) | A severe bark *D. Ponderosa* outbreak resulted in a 52%-60% | Highly dependent from the size of the forest studied | Count the number of trees killed by bark beetles until the end of the |

| | | | |
|---|---|---|---|
| | reduction in tree numbers at large landscape scale (>2000km²) (Morehouse et al., 2008; Pfeifer et al., 2011) In Wallonia and East France, *I. Typographus* outbreak resulted in 12.6% reduction of spruce forest area in 6 years (Arthur, G., et al. 2024). | but for a grid cell of 2500km2, ones could expect a [25%, 45%] reduction over the entire course of a massive outbreak. Use in the calculation of *score3* | outbreak, then divide by the number of trees just after the storm event. |
| Endemic mortality rate (endemic) | Total background mortality is around 1.2%.year$^{-1}$. Bark beetles as a functional group are estimated to account for 40% of the total mortality in the United States ($\approx$0.5%.year$^{-1}$) (Berner et al., 2017; Das et al., 2016; Hlásny et al., 2021b). | Not enough data was available to estimate a range. Nonetheless we decided to calculate a range including a 10% uncertainty [0.45-0.55] %.year$^{-1}$. Use in the calculation of *score4* | After the end of the outbreak, count the number of trees that die every year. Then average it. |


### 3.5.    Sensitivity to climate and windthrow

In this simulation experiment, the influx of fresh dead tree hosts ($N_{wood}$) used for bark beetle breeding was controlled
by modifying the maximum damage rate of a windthrow event ($DR_{windthrow}$) in ORCHIDEE. Seven $DR_{windthrow}$ were
simulated (i.e, 0.1%, 5%, 7.5%, 10%, 15%, 20%, 35%). Given the monotonic nature of the relationships between
$DR_{windthrow}$ and $i_{hosts\ dead}$ (Eq. 12), each event triggers a proportional increase in the dead host availability ($i_{hosts\ dead}$)
scaling between 0 and 1 (Fig. 3). Through its equations, ORCHIDEE assumes that for damage rates above 20% the
variable $i_{hosts\ dead}$ ($N_{wood}$) will always be equal to 1.0. $i_{rd\ spruce}$, however, may further decrease with increasing windthrow
damage, which makes the 35% damage rate still interesting to investigate. Although the simulations were run for all
$DR_{windthrow}$, only four windthrow damage rates including a windstorm resulting in  a 35% damage rate (Fig. 3), were
presented to enhance the readability of the result section.

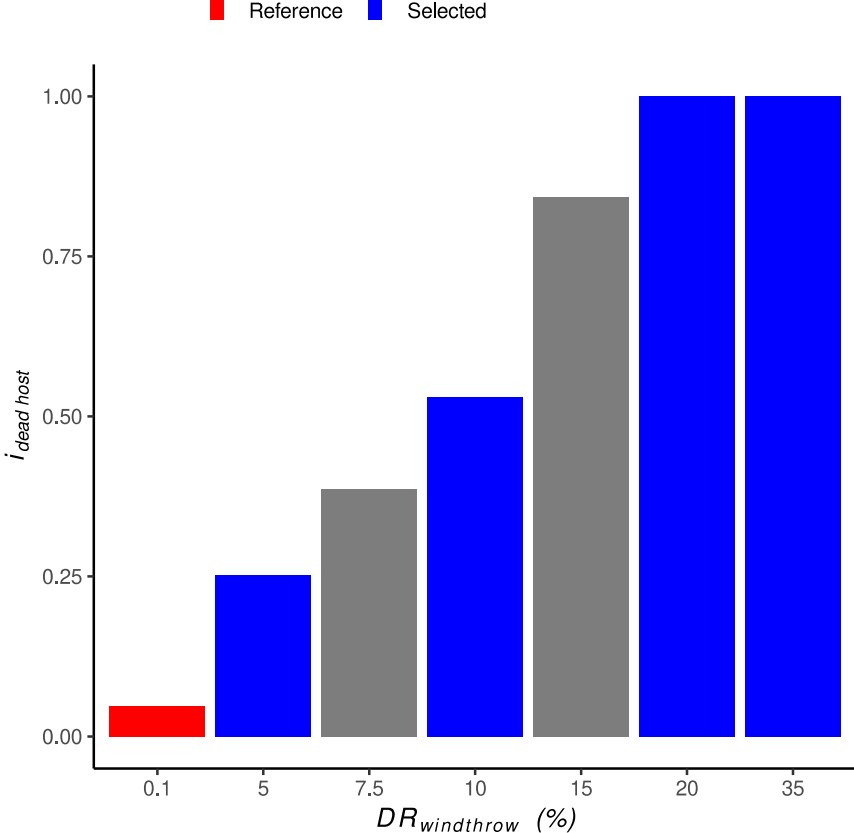

**Figure 3: Relationship between windthrow damage rate ($DR_{windthrow}$) and dead host index ($i_{hosts\ dead}$). For each site a $DR_{windthrow}$= 0.1% was used as the reference simulation because an endemic bark beetle population is expected following such a low intensity windthrow event. The four $DR_{windthrow}$ shown in blue were selected for subsequent presentation of the results because they cover the entire range for the $i_{hosts\ dead}$.**


The main driver of the number of generations a bark beetle population can achieve in one year is the number of days
higher than 8.3°C during winter time (Temperli et al., 2013) which is the reason why temperature is so important for
bark beetle reproduction. By taking REN, THA, WET and HES, the number of bark beetle generations ranged from
0.8 to 3.5 (Fig. 4) which is similar to the number of generations observed across Europe (Faccoli and Stergulc, 2006;
Jönsson et al., 2009, 2011). Limiting the analysis to only four sites simplifies the presentation without affecting the
range under investigation.

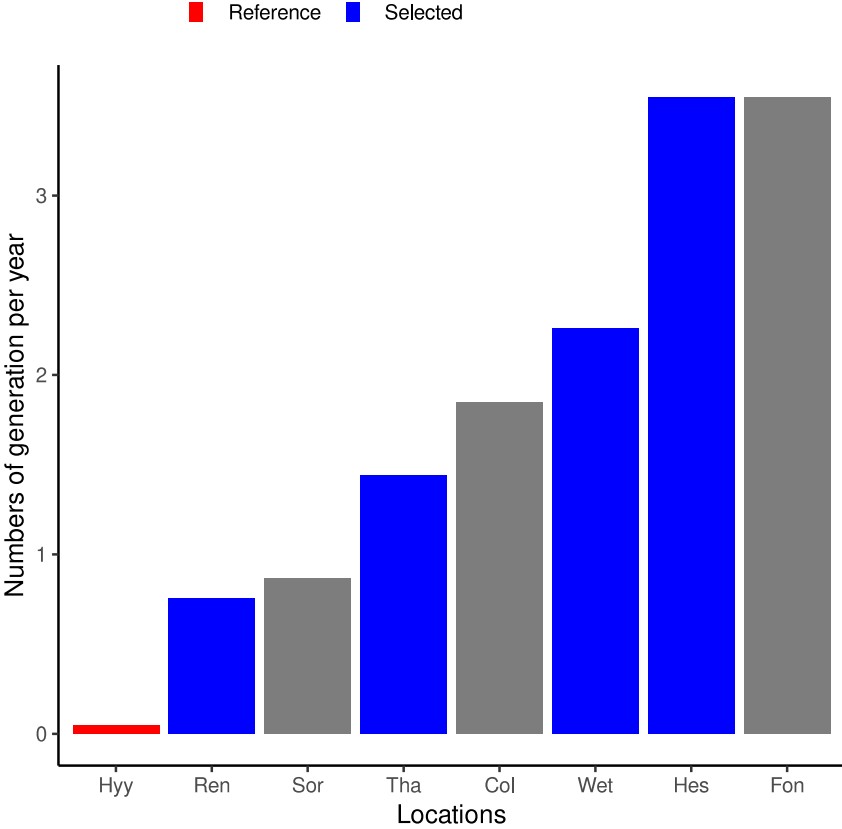

**Figure 4: Average number of bark beetle generations during the 5 years following the wind storm for at eight locations along a climate gradient. The HYY location in Finland was selected as the reference for the REN, THA, WET and HES locations. Only results from the reference and four selected locations (shown in blue) are shown in the results to enhance readability.**


For the climate gradient, the simulation for HYY served as a reference since the number of generations is lower than
1 for which no outbreak should happen under any circumstances. Under present climate conditions, an outbreak in
HYY should be considered an undesirable model result. Likewise, a $DR_{windthrow}=0.1\%$ is considered too low to trigger
an outbreak and was therefore used as the reference for the wind damage rate tests.

The experiment consisted of 40 simulations, i.e., 8 sites (including the reference) x 5 wind damage rates (including
the reference). Although the simulations were also run for SOR, COL and FON their results were found to be too
similar to the results of selected sites to present them as well. Hence, the result section presents only 25 out of the 40
simulations. Three output variables were assessed: bark beetle damage rate ($DR_{beetles}$), total biomass ($B_{total}$), and net
primary production ($NPP$). Total biomass was investigated over 100 years whereas $DR_{beetles}$ and $NPP$ were assessed
for the first 20 years following a windthrow.

### 3.6.    Continuous vs abrupt mortality

Where most land surface models use a fixed turnover time to simulate continuous mortality (Pugh et al., 2017; Thurner et al., 2017), ecological reality is better described by abrupt mortality events. An idealized simulation experiment was used to qualify the impact of abrupt mortality on net biome productivity by changing from a framework in which mortality is approximated by a constant background mortality to a framework in which mortality occurs in abrupt, discrete events. The impact of a change in mortality framework was assessed with an idealized simulation experiment that compares three configurations of ORCHIDEE: (1) a configuration that simulates mortality as a continuous process, labeled "the continuous configuration" which corresponds to previous versions of ORCHIDEE, and (2) a configuration capable of simulating abrupt mortality from windthrow and subsequent bark beetle outbreaks, labeled "the abrupt configuration" and (3) a configuration in which windthrow is activated but bark beetles outbreak is implicitly accounted for in the background mortality. This third configuration enabled attributing the impact to windthrow. The effect of simulating abrupt mortality was evaluated over 20, 50, and 100 year time horizons.

The impact of changing the mortality framework from continuous to abrupt was quantified on the basis of 120 simulations (8 locations x 7 windthrow damage rates x 2 configurations + 8 sites x 1 configuration) of 100 years each.

The simulations with abrupt mortality were run first. Subsequently, the number of trees killed was quantified and used as a reference value for the continuous mortality set-up. This approach resulted in the same quantities of dead trees at the end of the simulation for both frameworks, which then differed only in the timing of the simulated mortality. This precaution is necessary to avoid comparing two different mortality regimes where the result would mainly be explained by the intensity of the mortality rather than by its underlying mechanisms.

Changes in forest functioning were evaluated through the temporal evolution of accumulated net biome productivity (*NBP*) over a 100-years time frame. *NBP* is defined as the regional net carbon accumulation after considering losses of carbon from fire, harvest, and other episodic disturbances. In ORCHIDEE, *NBP* is calculated following the definition by Chapin et al. (2006) as the carbon remaining in the biomass, litter and soil after accounting for photosynthesis, and respiration because fire, harvest, leaching and volatile emissions were not accounted for in this simulations experiment.

## 4.    Results
### 4.1.    Sensitivity to model parameter sets

The impact of spruce stand competition ($i_{hosts\ susceptibility}$) on outbreak dynamics was examined by adjusting the parameters $S_{susceptibility}$ and $i_{rd\ susceptibility}$ in equation 6a'. When $S_{susceptibility}$ resulted in a linear relationship ($S_{susceptibility}$ = -1.0), no peak in bark beetle damage occurred for the three tested values of $i_{rd\ susceptibility}$ (permissive, reference, restrictive) at a 10% windthrow damage rate (Fig. 5, panel h). However, employing a step function ($S_{susceptibilty}$ = -

*500.0*) led to either sporadic peaks of bark beetle damage with a permissive $i_{rd\ susceptibility}$ or a two-year outbreak with a
maximum damage rate of 60% with a restrictive $i_{rd\ susceptibility}$ (Fig. 5, panel h), neither of which aligns with the
observations summarized in Table 3.

The closest outcome to observation from table 3 was obtained with a logistic relationship ($S_{susceptibility}$ = -5.0), where
$i_{rd\ susceptibility}$ determined the duration of the outbreak: 11, 16, and 25 years for restrictive, reference, and permissive
parameter values, respectively (Fig. 5, panel h). Either the restrictive or reference parameter value could be utilized
since a range of 11-16 years aligns with the observations (Table 3). To examine the occurrence of improbable
outbreaks, sensitivity tests were repeated for a 0.1% windthrow damage rate. None of the nine parameter
combinations triggered an outbreak (Fig. 5, panel g), suggesting that improbable outbreaks  due to the calculation of
$i_{hosts\ susceptibility}$ are unlikely.

From the calculation of the credibility score, only one set obtains a score of 4 ($S_{susceptibility}$ = *-5.0, $i_{rd\ susceptibility}$=0.55*,
Table s1). The concerning parameters value has been selected and reported in table 4.

The effect of the capability of bark beetle to mass attack ($i_{beetles\ mass\ attack}$) when the population exceeds a threshold was
evaluated by varying $S_{mass\ attack}$ and $BP_{limit}$ (Eq. 14). Linear relationships ($S_{mass\ attack}$ = -1.0) resulted in similar outbreak
dynamics for all $BP_{limit}$ values, with the model settling on a constant endemic damage following an outbreak, though
higher than observed (Table 3, Fig. 5, panel f). Introducing a logistic or step function slightly altered outbreak
dynamics except when assuming a step function for the restrictive value, which prevented an outbreak. Repeating
sensitivity tests for a 0.1% windthrow damage rate showed that assuming linear or logistic relationships could trigger
an outbreak (Fig. 5, panel e), indicating that improbable outbreaks may arise from the calculation of $i_{hosts\ mass\ attack}$.

From the calculation of the credibility score, three sets obtain a score of 4 but only set 4.6 was chosen because of its
intermediate position compared to sets 4.9 and 4.5 (Table s1). The concerning parameter values ($S_{mass\ attack}$ = *-30.0*,
$BP_{limit}$=*0.06*) have been selected and reported in table 4.

The impact of bark beetle activities from the previous year ($i_{beetles\ activity}$) on outbreak dynamics was investigated by
varying $S_{activity}$ and $act_{limit}$ (Eq. 10). Linear or logistic relationships resulted in excessively long outbreaks (>30 years)
compared to observations (Table 3, panel b), whereas assuming a step-function relationship simulated a decline in
the outbreak after 14 years. Sensitivity tests repeated for a 0.1% windthrow damage rate showed that assuming a
linear relationship could trigger an improbable outbreak (Fig. 5, panel a) through the calculation of $i_{beetles\ activity}$.

From the calculation of the credibility score, only one set obtains a score of 4 ($S_{activity}$ = *-500.0, $act_{limit}$=0.12*,  Table
s1). The concerning parameters value has been selected and reported in table 4.

To explore the effect of the numbers of generation ($i_{beetles\ generation}$) on the outbreak dynamics, $S_{generation}$ and $G_{limit}$ from
equation 9 were varied. Bark beetle damage rate was more sensitive to $G_{limit}$ than $S_{generation}$, but only a linear
relationship with the reference $G_{limit}$ = 1.0 yielded an intermediate outbreak intensity consistent with the continental
climate at the test location (i.e., THA, Fig. 5, panel d). Other combinations resulted in either too strong or no peak
during the outbreak. Repeating sensitivity tests for a 0.1% windthrow damage rate showed that none of the nine
parameter combinations triggered an outbreak (Fig. 5 panel c), indicating that improbable outbreaks from the
calculation of $i_{beetles\ generation}$ are unlikely.

From the calculation of the credibility score, three sets obtain a score of 4 but only set 1.4 was chosen because of its
intermediate position compared to sets 1.1 and 1.5 (Table s1). The concerning parameter values ($S_{generation}$ = *1.0*,
$G_{limit}$=*1.0*) have been selected and reported in table 4.

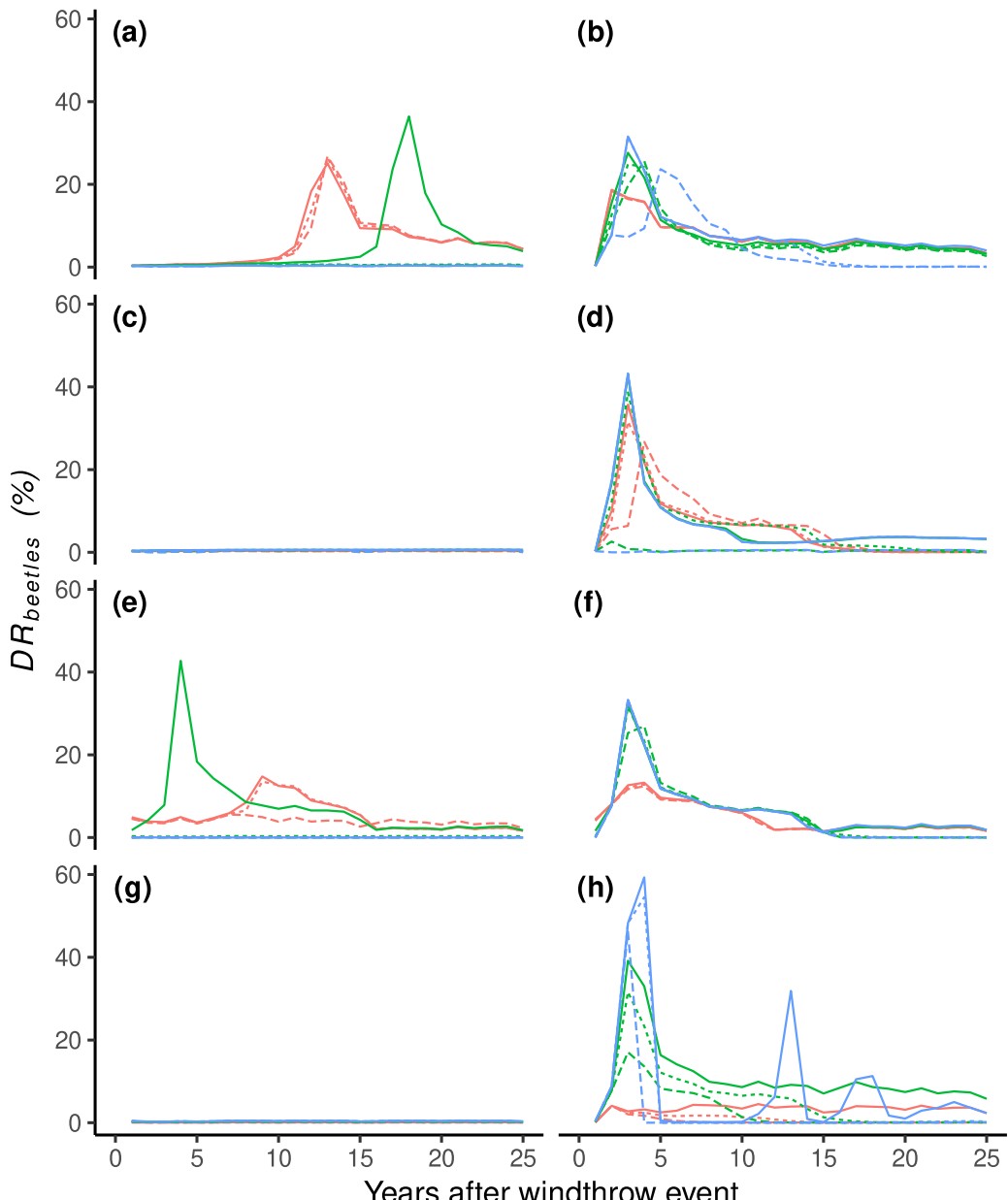

**Figure 5: Simulation results from the sensitivity experiment at the THA site. Eight parameters from four equations were evaluated. Each equation represents an index from the bark beetle outbreak model ($i_{hosts\ susceptibility}$ $i_{hosts\ mass\ attack}$, $i_{beetles\ activity}$, $i_{beetles\ generation}$). Each index is represented by a logistic function defined by a shape parameter (*Shape*) and a limit parameter (*Limit*). Three values were chosen for each parameter resulting in 9 pairs of parameters for each index. Colored lines represent the shape parameter varying from linear : *Shape* = -1.0 (red), logistic -5.0 < *Shape* < -30.0 (green), to step function where *Shape* = -500.0 (blue). Line type represents three different values for *Limit* parameters where references (dashed line) are values of $i_{rd\ susceptibility}$, $BP_{limt}$, $act_{limit}$ and $G_{limit}$ (given in table 4), whereas permissive (full line) and restrictive (dashed dotted) represent a 50% decrease or increase respectively.**


**4.2. Model tuning**
By comparing the outcomes of the sensitivity tests (section 4.1) to a compilation of observations (Table 3), a first
estimate for several parameters was proposed (Table 4).

**Table 4: Parameter values from the bark beetle model based on the score obtained in the sensitivity analysis. (*) parameter values deliberately fixed and excluded from the sensitivity analysis (section 3.3 for justification).**

| Parameter | Source | Chosen parameters |
|---|---|---|
| $S_{generation}$ | This study: from SA (see 3.1.4) | -1.0 |
| $G_{limit}$ | Adapted from Temperli et al. 2013 | 1.0 |
| $DD_{ref}$ | Adapted from Temperli et al. 2013 | 547.0 (*) |
| $S_{drought}$ | Adapted from Temperli et al. 2013 | -9.5 (*) |
| $PWS_{limit}$ | Adapted from Temperli et al. 2013 | 0.4 (*) |
| $max_{Nwood}$ | This study: scale dependent (see 2.4.2) | 0.2 (*) |
| $S_{activity}$ | This study: from SA (see 3.1.3) | -500.0 |
| $act_{limit}$ | This study: from SA (see 3.1.3) | 0.06 |
| $S_{susceptibility}$ | This study: from SA (see 3.1.1) | -20.0 |
| $i_{rd\ susceptibility}$ | This study: from SA (see 3.1.1) | 0.55 |
| $S_{competition}$ | This study: from SA (see 3.1.1) | -5.0 (*) |
| $i_{rd\ limit}$ | This study: scale dependent (see 2.4.1) | 0.4 (*) |
| $S_{mass\ attack}$ | This study: From SA (see 3.1.2) | -30.0 |
| $BP_{limit}$ | This study: scale dependent (see 3.1.2) | 0.12 |
| $S_{share}$ | This study: not used (see 2.5) | 15.5 (*) |
| $SH_{limit}$ | This study: not used (see 2.5) | 0.6 (*) |


**4.3. Impact of climate and windthrow on bark beetle damage**
In ORCHIDEE, the warmest sites, HES and WET, experienced significant bark beetle outbreaks across a wide
spectrum of windthrow mortality rates, whereas colder sites like REN and THA saw outbreaks only in response to
the most severe windthrow events (Fig. 6, panel b, c). A greater average number of bark beetle generations in the
years following windthrow events led to higher bark beetle damage rates at the peak of outbreaks. For instance, at a
35% windthrow mortality rate, HES reached a maximum bark beetle damage rate of 50%, whereas REN's maximum
was 22% (Fig. 6 panel a, b).

Interestingly, high tree mortality rates from windthrow could also lead to delays and lower maximum $DR_{beetles}$ (Fig.
6). For instance, at the HES site, 10%, 20%, and 35% windthrow damage rates triggered maximum $DR_{beetles}$ of 50%,
43%, and 37%, respectively (Fig. 6 panel a). Conversely, low $DR_{windthrow}$, like 5% at WET, delayed the peak of bark
beetle outbreaks by 9 years (Fig. 6, panel d). Additionally, the model simulated a post-epidemic stage during which
the outbreak damage rate remained relatively low (<10%) and lasted between 3 to 10 years (Fig. 6). Overall, the
simulated outbreaks lasted between 11 to 20 years, consistent with field observations (Table 3).

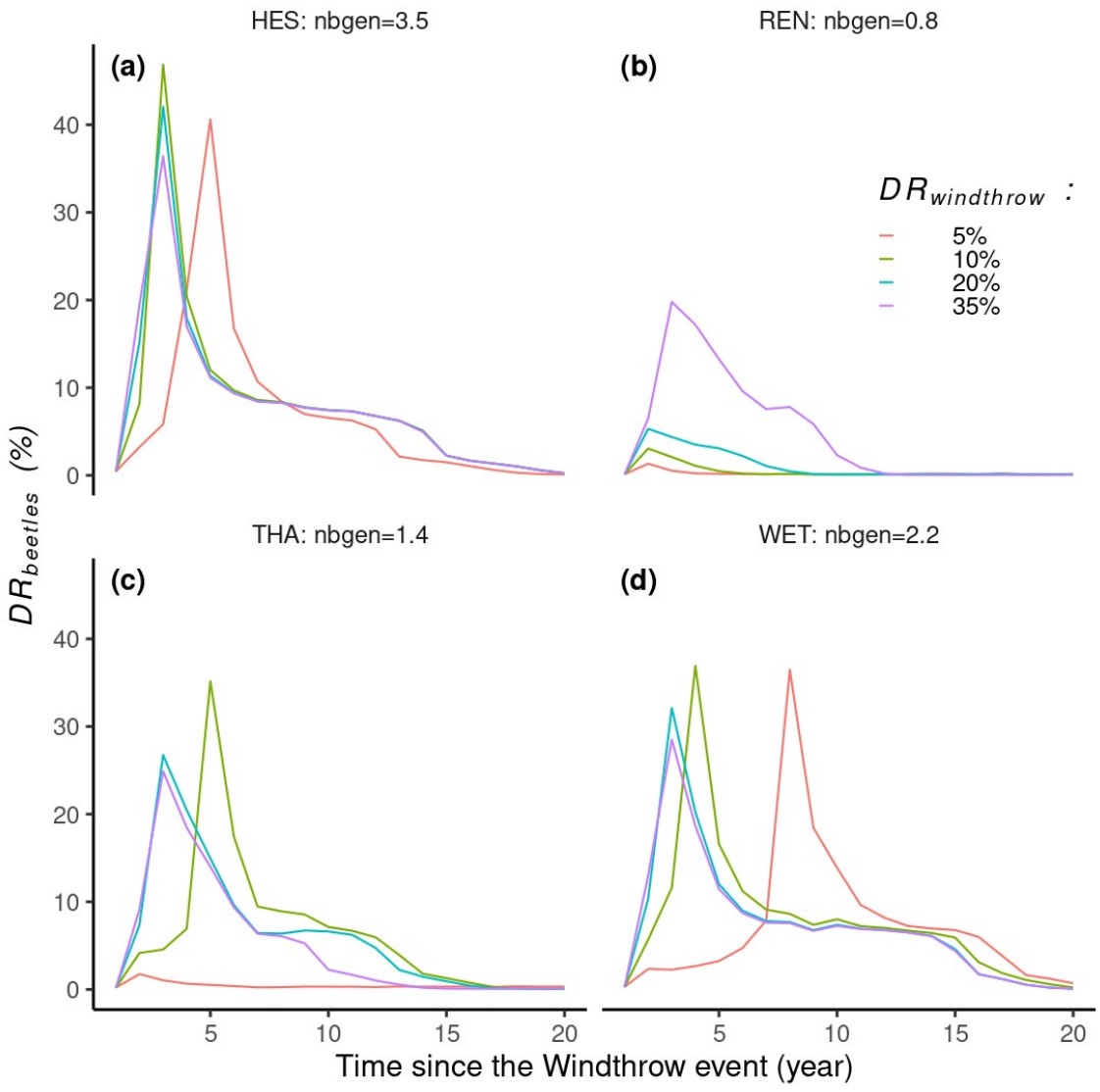

**Figure 6: Simulation results of 16 simulations (4 locations x 4 windthrow damage rates $DR_{windthrow}$). Lines represent the annual bark beetle damage rate as a fraction of the total biomass ($DR_{beetles}$). *Nbgen* is the average number of bark beetle generations during five years after the windthrow event. $DR_{windthrow}$ represents the percentage of biomass loss by a windthrow event at the start of the simulation.**


At the coldest site, HYY, ORCHIDEE simulated only a small number of bark beetle generations, preventing
outbreaks from occurring. This observation validates the initial parameter tuning (Table 4), indicating that it is
robust enough to prevent improbable outbreaks, such as the model triggering outbreaks in sites where bark beetles
cannot reproduce.

**4.4. Impact of climate and windthrow on stand biomass and Net Primary Production**

All locations experienced a 10 to 20 years decrease in total biomass until at most 9 kgC.m$^{-2}$ at which time the
outbreak ended (Fig. 7, panel a, b, c, d). The model can simulate significant epidemic events even if the initial
trigger, such as the windthrow event in our study, is not particularly intense. Once the bark beetles can mass attack
living trees, the bark beetle population ($i_{beetles\ pressure}$) will increase and kill more and more trees until so many trees are
killed that the stand density of the remaining living trees drops below the threshold of $i_{rd\ spruce} = i_{rd\ limit}=0.4$. In
ORCHIDEE, an $i_{rd\ limit}=0.4$ for spruce forest corresponds to a biomass of around 9 kgC.m$^{-2}$ which in ORCHIDEE is
too low to maintain an epidemic population of bark beetles at the 2500 km2 grid cell. Interestingly, for the climate
observed at REN where the number of generations is approximately one, the bark beetle population   can only
become epidemic t following an intense windthrow event  with a 35% damage rate (Fig. 7).

Throughout the outbreak period, there was a notable decrease in net primary production (*NPP*)(Fig. 7). This decrease
is primarily attributed to a sharp decline in leaf area index (not shown). Following the epidemic phase, the leaf area
recovers. Following the outbreak: the reduction in stand tree density due to bark beetle damage decreases autotrophic
respiration   (not shown) and the sparser canopy allows more light to reach the forest floor where it fosters
recruitment (not shown), resulting a higher *NPP* or forest growth (Fig. 7). Consequently, carbon use efficiency tends
to be higher in sparsely populated stands compared to densely populated ones.

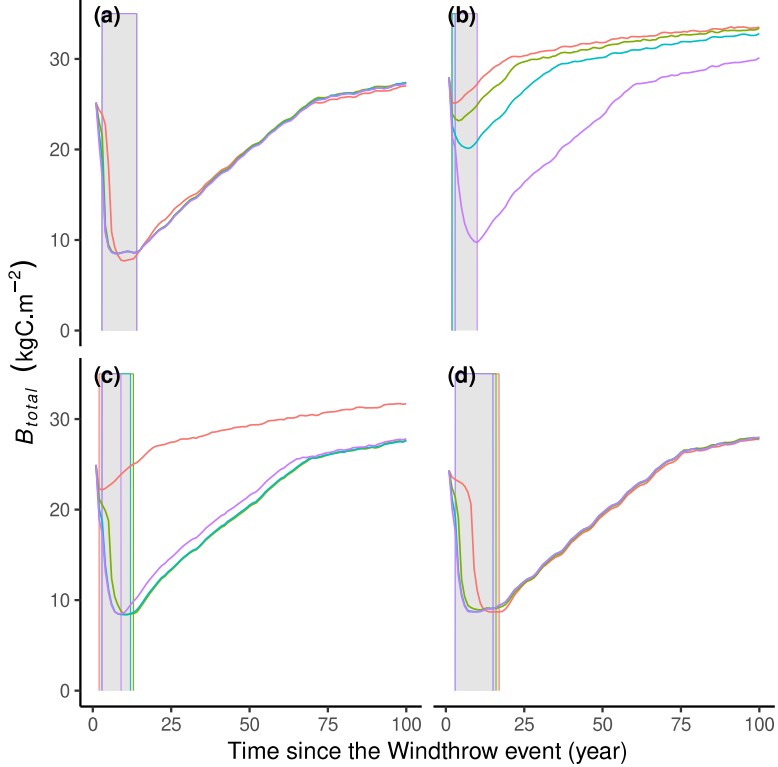

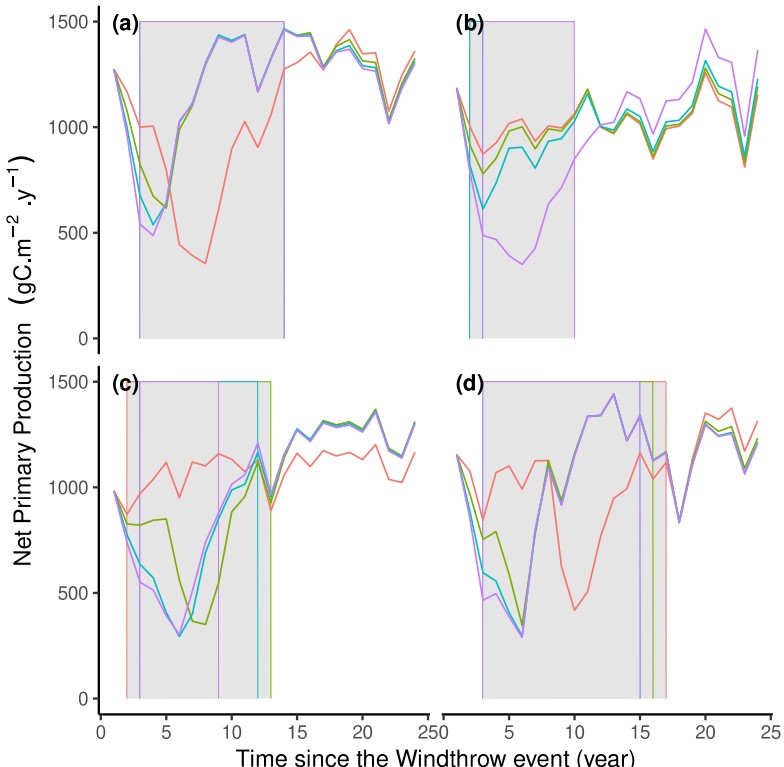

**Figure 7: Simulation results of 16 simulations (4 sites x 4 windthrow mortality rate). Lines represent the annual average net primary production (NPP) in gC.m$^{-2}$.y$^{-1}$ or total stand biomass ($B_{total}$) in kgC.m$^{-2}$. *Nbgen* is the average number of bark beetle generations during the five years after the windthrow event. *DR$_{windthrow}$* represents the percentage of biomass loss by a windthrow event at the start of the simulation. Grey areas represent the epidemic phase.**


**4.5. Continuous vs. abrupt mortality**


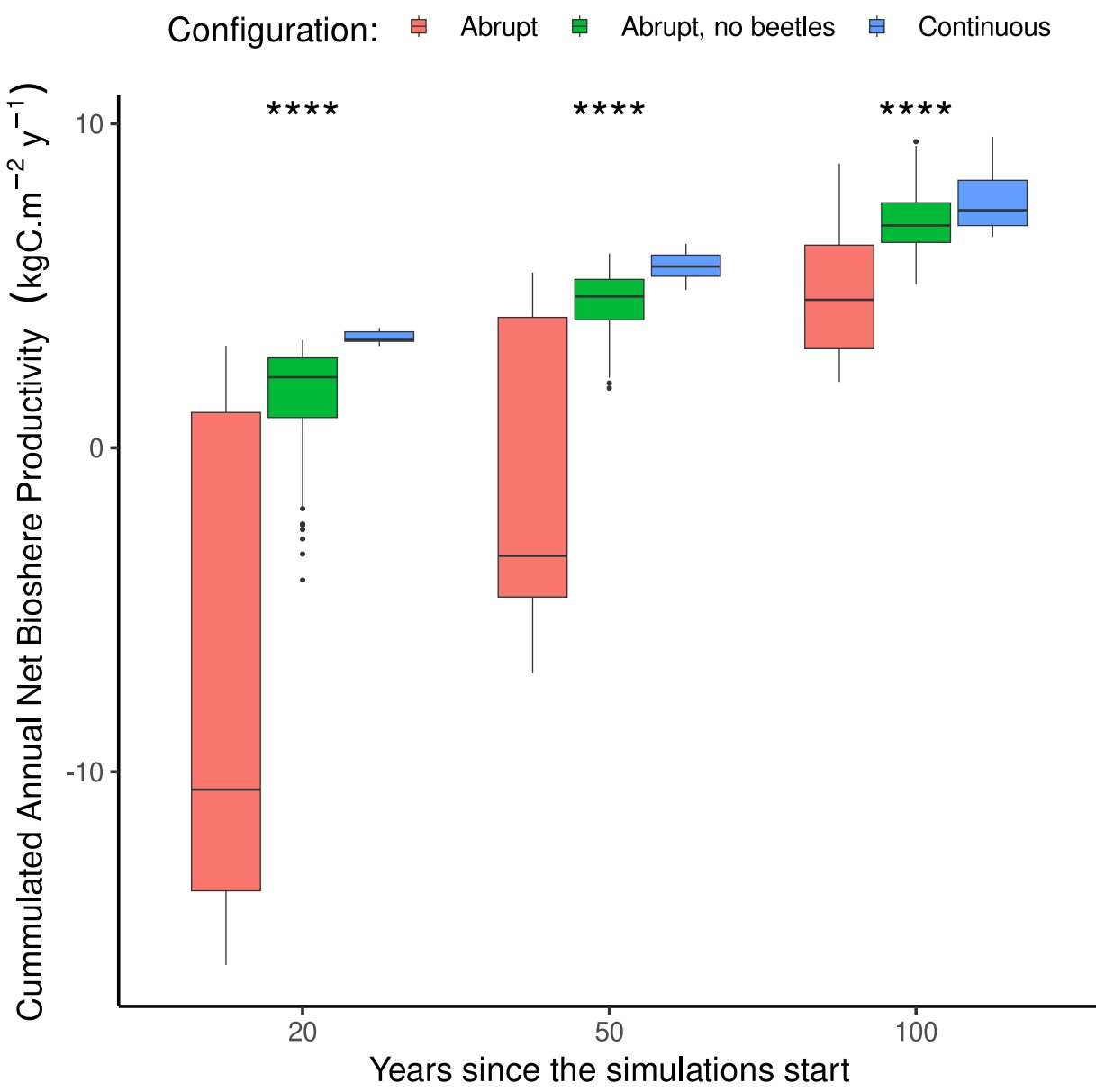

**Figure 8: Difference in cumulative net biome production at three discrete time horizons (i.e. 20, 50 and 100 years) between a fixed continuous mortality rate (blue, n=8), abrupt tree mortality from a windstorm and the subsequent bark beetle outbreak (red, n=56), abrupt mortality from a windstorm not followed by a bark beetles outbreak (green, n=56). Note that in the continuous mortality configuration the mortality rate was adjusted to obtain a similar number of trees killed after 100 years as in the abrupt mortality configuration. The variation of each boxplot arises due to different locations and prescribed storm intensities. Each boxplot displays the median value (thick horizontal line), the quartile range (box border), and the 95% confidence interval (vertical line). A Wilcoxon test between the three configurations at each time horizon showed significant differences (p-value<0001) denoted by the four stars.**

The total accumulated net biome production (*NBP*) was evaluated using the ORCHIDEE model across three
different timeframes: 20, 50, and 100 years. At the 20-years mark, the average accumulated *NBP* notably differed
between the 'continuous', 'abrupt' and the abrupt without bark beetles outbreak ('no beetles') mortality
configurations: -7.12±0.97, -1.37±0.28 and 3.39±0.74 kgC.m$^{-2}$.y$^{-1}$ for the 'abrupt', 'no beetles' and 'continuous'
mortality configurations, respectively. These differences were statistically significant (Wilcoxon, p-value<0001),
indicating a substantial initial reduction in *NBP* with the 'abrupt' configurations, as ecosystems behaved as carbon
sources, whereas under the 'continuous' configuration, they acted as carbon sinks (Fig. 8). The variability in *NBP*
demonstrated the broad temperature gradient in Europe and indicated that despite many locations potentially acting
as sources under the 'abrupt' configuration, some may transition to carbon sinks within the first 20 years following a
disturbance.

Moving to the 50-years horizon, the difference between the three frameworks decreased, with net biome productions
of -0.81±0.60, 4.43±0.15 and 5.61±0.18 kgC.m$^{-2}$.y$^{-1}$ for the 'abrupt', 'no beetles' and 'continuous' mortality
configuration, respectively. The difference in sink strength remained statistically significant (Wilcoxon, p-
value<0.001), with the *NBP* in the 'abrupt' configuration approaching carbon neutrality while without the
consecutive bark beetles outbreak the ecosystems already became a carbon sink. The climate conditions had a lasting
effect on the responses, with the 'abrupt' configuration showing a greater range in responses compared to the
'continuous' one.

At the 100-years mark, the average cumulative *NBP* for the 'abrupts' and 'continuous' configurations approached
each other with values of 4.85±0.26, 7.09±0.17 and 7.73±0.40 kgC.m$^{-2}$.y$^{-1}$, respectively (Fig. 8) but were still
significantly different (Wilcoxon, p-value<0.001). ORCHIDEE simulated a return to a carbon sink (indicated by
positive cumulative *NBP* values) suggesting a long-term recovery and potential return to pre-disturbance
productivity levels within a century following the windthrow and beetle outbreak event. The 'continuous'
configuration displayed a consistently higher median value, suggesting weaker impact of tree mortality dynamics on
the long term carbon cycle.

**5. Discussion**
**5.1. Simulating the dynamics of bark beetle outbreaks and their interaction with windthrow**
Our *Ips typographus* outbreak model has demonstrated its capability to simulate a broad range of disturbance
dynamics. The variation in the outbreak dynamics and the response of the outbreak to its main drivers (Fig. 5 & 6)
give confidence in the ability of ORCHIDEE to simulate various outbreak scenarios observed across the temperate
and boreal zones under changing climate conditions.

Windthrow events have significant ecological impact because such disturbances offer fresh breeding substrates,
which in turn increase bark beetle populations (Lausch et al., 2011). Our model results align with these findings,
indicating that windthrows causing damage of 5% or more may trigger beetle outbreaks (Fig. 6). Additionally, a
strong increase in bark beetle populations has been observed following a windthrow event (Wermelinger, 2004), a
pattern reflected in the ORCHIDEE simulations. The model simulates a buildup stage spanning 1 to 9 years, where
bark beetle numbers increase prior to peaking, with the duration influenced by the severity of the windthrow and the
prevailing climate (Fig. 6).

Temperature is another critical factor affecting bark beetle life cycles. Intra- and interannual variation in temperature
impact bark beetles, with warmer conditions fostering multiple generations per year, whereas cooler, damp climates
slow breeding and survival rates (Benz et al., 2005). In line with these findings, the temperature dependence of the
ORCHIDEE simulations show that cold winters at locations such as SOR and REN reduced bark beetle activity
compared to warmer locations like THA and WET (Fig. 6). Lieutier et al. (2004) documented that if the population
is large enough, bark beetles can mass attack healthy trees. Our model incorporates this dynamic, illustrated by
epidemic stages where living trees become viable hosts, which then exacerbates the growth of the beetle population
(Fig. 1).

The aftermath of a windthrow and subsequent bark beetle outbreak also affects the forest carbon and nitrogen cycles.
This impact is observed in the form of snags which are standing dead trees that undergo decomposition. Snags can
temporarily disrupt the link between soil and ecosystem carbon and nitrogen dynamics (Rhoades, 2019; Custer et al.,
2020). While in ORCHIDEE, the decay of fallen logs does not account for snags yet, the model suggests a recovery
period ranging from 5 to 15 years, contingent upon the intensity of the bark beetle outbreak (Fig. 7). As snags create
gaps in the canopy, conditions favorable to natural forest regeneration emerge (Jonášová and Prach, 2004) .

**5.2.**       **Emerging properties from interacting disturbances**

While this study did not precisely quantified the impact of simulating  abrupt mortality rather than approaching
mortality as a continuous process, it demonstrated that the impact of abrupt mortality varies across location and time,
i.e. ecosystem functions, such as carbon storage, are affected by natural disturbances like *Ips typographus* outbreaks,
having significant impacts on short to mid-term carbon balance estimates (Fig. 8). The simulation experiments also
highlighted that the legacy effects of disturbances can endure for decades; even for a simplified representation of
forest ecosystems such as ORCHIDEE, where the recovery might be too fast due to the absence of snags (Senf et al.,

2017).


The ability to simulate resistance (i.e., staying essentially unchanged despite the presence of disturbances; Grimm
and Wissel, 1997) as an emerging property is evident from Figs. 6 and 7 for locations REN, where no bark beetle
outbreaks were observed following a medium windthrow event (5%-20%). However, in all simulated locations that
could not resist a bark beetle outbreak, the forest was resilient (i.e., returning to the reference state or dynamic after a
temporary disturbance; Grimm and Wissel, 1997) and ecosystem functions were restored to the level from before the
windthrow. The elasticity (the speed of return to the reference state or dynamic after a temporary disturbance;
Grimm and Wissel, 1997) of the carbon sink capacity ranged from 7 to 14 years. This elasticity is in line with the
little observational evidence of ecosystem shifts due to natural disturbances in forests (Millar and Stephenson, 2015).
Finally, after the disturbance and the recovery of vegetation structure, the ecosystems simulated by ORCHIDEE
showed persistence (i.e. continue along their initial developmental path; Grimm and Wissel, 1997).

**5.3.**       **Are cascading disturbances important for carbon balance estimates ?**
The enhanced complexity introduced into the ORCHIDEE model by incorporating abrupt mortality events, as
opposed to a continuous mortality, prompts the question: does this model refinement yield new insights into carbon
balance estimates? Our century-long analysis demonstrated that the net biome production, a the metric for carbon
sequestration, ultimately converges between the continuous and abrupt mortality frameworks (Fig. 8). This suggests
that irrespective of the nature of the mortality events, the forest ecosystem goes through a recovery phase, marked by
increased growth that compensates for the growth deficits during the disturbance.

Yet, our experiment has not taken into account the frequency of disturbances. Given the profound influence of
disturbance legacies on carbon dynamics, a recurrence interval shorter than the recovery time of the forest might
result in a tipping point. Such a scenario could diminish the carbon sequestration potential of the forest beyond 100-
year timeframe, and in extreme cases, may even lead to ecosystem collapse, outcomes not explored in the current
simulations nor documented in the recent literature (Millar and Stephenson, 2015).

In the mid-term, spanning 20 to 50 years, the widely used continuous mortality model appears to inflate the carbon
sink capabilities of forests when juxtaposed with abrupt mortality scenarios. Since policy frameworks, including the
Green Deal for Europe (2023) and the Paris Agreement |(UNFCCC, 2023), upon these medium-term predictions,
they would benefit from adopting model simulations that integrate abrupt mortality events to avoid an
overestimation of carbon sink capacities of forest. Furthermore, the accuracy of carbon balance estimates strongly
depends upon the initial state of the forest in the model. Forest conditions markedly affect carbon uptake rates. Thus,
incorporating an abrupt mortality framework into the ORCHIDEE model could substantially refine and strengthen
the predictive power of our carbon balance assessments across short, medium, and long-term scales.

**5.4.**    **Shortcomings of the bark beetle outbreak model**
The bark beetle outbreak model developed in this study builds upon the strengths of the previously established
LandClim model, though it also inherited some of its limitations. One notable shortcoming is the model for bark
beetle phenology, which is an empirical model making use of accumulated degrees-days. Since the conception of the
phenology model a decade ago, Europe's climate has undergone substantial changes, primarily manifested in warmer
winters and springs (Copernicus, 2024). Because of these changes, chances have increased for two or even more
bark beetle generations within a calendar year (Hlásny et al., 2021a). These changes call for an update of the beetle's
phenology model to align with these more recent observations (Ogris et al., 2019).

A second limitation is that our study, ORCHIDEE, has been parameterized to simulate only *Ips typographus* in
Europe. In order to change the beetles and tree host interactions e.g. pine bark beetle in North America
(*Dendroctonus monticolae Hopkins*), the sensitivity of indexes must be revised, for example, pine beetle is not
breeding on the dead wood falling from withrow but very sensitive to drought events (Preisler et al., 2012). $i_{hosts\ defense}$,
and $i_{hosts\ dead}$ as well as the phenology model will need to be revised.

Another issue is the model's consideration of drought. As outlined in the method section, drought is treated as an
exacerbating factor, rather than a primary trigger as is the case for windthrow. This understanding was accurate for
*Ips typographus* a decade ago (Temperli et al., 2013); however, emerging evidence increasingly suggests that
drought events may indeed trigger bark beetle outbreaks across Europe (Nardi et al., 2023; Netherer et al., 2015).
Consequently, this extreme drought as a trigger should be incorporated in a future revision of ORCHIDEE's *Ips*
*typographus* outbreak model.

**6.    Outlook**
This study simulated the one-way interaction between windthrow and *Ips typographus* outbreaks in unmanaged
forests. Future research will incorporate additional interactions, such as: the interplay between droughts, storms, and
bark beetles; storms, bark beetles, and fires; as well as forest management, storms, and bark beetles.

The bark beetle outbreak model could also be enhanced by simulating: (a) standing dead trees (or snags), which
would help account for differences in wood decomposition between snags and logs (Angers et al., 2012; Storaunet et
al., 2005), (b) the migration of bark beetles to neighboring locations, which becomes significant to account for in a
model that operates at spatial resolutions below approximately 10 kilometers, and (c) an up-to-date beetle phenology
model which accounts for the recent change in their behavior induced by climate change.

This research provided an initial qualitative assessment of a new model feature. However, the application of the
model necessitates an evaluation of the simulations against observations of cascading disturbances at the regional
scale, which is the topic of an ongoing study.

**7.    Conclusion**
Our approach enables improving the realism of the *Ips typographus* model in ORCHIDEE without reducing its
generality (Levins, 1966). The integration of a bark beetle outbreak model in interaction with other natural
disturbance such as windthrow into the ORCHIDEE land surface model has resulted in a broader range of
disturbance dynamics and has demonstrated the importance to simulate various disturbance interaction scenarios
under different climatic conditions. Incorporating abrupt mortality events instead of a fixed continuous mortality
calculation provided new insights into carbon balance estimates. The study showed that the continuous mortality
framework, which is commonly used in the land-surface modeling community, tends to overestimate the carbon sink
capacity of forests in the 20 to 50 year range in ecosystems under high disturbance pressure, compared to scenarios
with abrupt mortality events.

Apart from these advances, the study revealed possible shortcomings in the bark beetle outbreak model including the need to update the beetle's phenology model to reflect recent climate changes, and the need to consider extreme drought as a trigger for bark beetle outbreaks in line with emerging evidence. Looking ahead, future work will further develop the capability of ORCHIDEE to simulate interacting disturbances such as the interplay between extreme droughts, storms, and bark beetles, and between storms, bark beetles, and fires.

The final step will be a quantitative evaluation based on observed data (Marini et al., 2017) in order to assess the capability of ORCHIDEE to simulate complex interaction between multiple sources of tree mortality affecting the carbon balance at large scale.

## 8. Code availability

- R script and data are available at :
  https://doi.org/10.5281/zenodo.12806280
- ORCHIDEE rev 7791 code is also available from:
  https://forge.ipsl.jussieu.fr/orchidee/browser/branches/publications/
ORCHIDEE_Bark_beetles_outbreak_gmd_2024

## 9. Data availability

- The Fluxnet climate forcing data are available at  https://fluxnet.org/
- The simulation results use in this study are available at https://doi.org/10.5281/zenodo.12806280

## 10. Author contribution

G. Marie, S. Luyssaert designed the experiments and G. Marie conducted them. Following discussions with H. Jactel, G. Petter and M. Cailleret, G. Marie developed the bark beetles model code and performed the simulations. J. Jeong integrated the wind damage and bark beetle models with each other. G. Marie, J. Jeong, V. Bastrikov, J. Ghattas, B. Guenet, A.S. Lansø, M.J. McGrath, K. Naudts, A. Valade, C. Yue, and S. Luyssaert, contributed to the development, parameterization and evaluation of the ORCHIDEE revision used in this study. G. Marie, J. Jeong, and S. Luyssaert prepared the manuscript with contributions from all co-authors.

## 11. Competing interests

No competing interest

## 12. Acknowledgements

GM was funded by MSCF (CLIMPRO) and ADEME (DIPROG). SL and KN were funded by Horizon 2020, HoliSoils (SEP-210673589) and Horizon Europe INFORMA (101060309). JJ and BG were funded by Horizon 2020, HoliSoils (SEP-210673589). GP acknowledges funding by the Swiss National Science Foundation (SNF

163250). ASL was funded by Horizon 2020, Crescendo (641816). C.Y. was funded by the National Science
Foundation of China (U20A2090 and 41971132). MJM was supported by the European Commission, Horizon 2020
Framework Programme (VERIFY, grant no. 776810) and the European Union's Horizon 2020 research and
innovation programme under Grant Agreement No. 958927 (CoCO2). AV acknowledges funding by Agropolis
Fondation (2101-048). This work was performed using HPC resources from GENCI-TGCC (Grant 2022-06328).
The Textual AI - Open AI GPT4 (https://chat.openai.com/) has been used for language editing at an early stage of
manuscript preparation.

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
