# Peer review of "Simulating *Ips typographus (L.)*, outbreak dynamics and their influence on carbon balance estimates with ORCHIDEE r8627"

_EGUsphere, 2023_

## Referee Comment (RC2)

**Review: Simulating Bark Beetle Outbreak Dynamics and their Influence on Carbon Balance Estimates with ORCHIDEE r7791**

In this manuscript, Marie et al. describe the implementation of a bark beetle outbreak model into the ORCHIDEE dynamic global vegetation model. Pests are an important driver of forest dynamics but are notoriously hard to simulate, so development in this area is always welcome. The authors do not attempt to reproduce specific real-world bark beetle outbreaks in this work, focusing instead on the model description and a more general evaluation of the model's behavior. They show that at least some patterns seen with regard to bark beetle dynamics, such as the duration of outbreaks and the effects of temperature, are reproduced well. I consider this effort a good first step and, in principle, worthy of publication.

However, there are a number of issues that prevent the current version of this manuscript from being accepted for publication. Some parts of the model description are confusing or too vague, and some analyses are insufficiently described (as I describe in my Substantive Comments below). The paper would also benefit greatly from some reorganization, and I have a large number of more minor comments and corrections. With all that in mind, I recommend this paper be **reconsidered after major revisions**.

**Substantive Comments**

1) Many symbols are missing from Table 1 (which should be moved to an Appendix). I've listed some but not all of them in the "minor comments/corrections" section, but please do a thorough review and add any others you find.

2) The references to beetle "generations" are hard to understand. They don't seem to fit with how beetle dynamics are ultimately realized in the model, but even in context here it's confusing. The sentence at lines 240-241 says, "the index G reaches its maximum value of one when 2.5 or more generations occur in a single growing season." But isn't G "the number of beetle generations… that could occur in the current year"? Then the next sentence is confusing because of "of the first generation." So far we've only been learning about beetle pressure index, not what actually triggers a generation.

3) It's really important to include something like Sect. 2.2.4 (differences from beetles in LandClim; lines 365-374), but it's currently too vague. The "calculation of the susceptibility" differences are documented (although not explained, per se) in Table S1, but that's not referenced in the text here. As far as I can tell, the other mentioned differences are not explained at all. A model description paper in *GMD*, as outlined in the Manuscript Types webpage, "should be sufficiently detailed to in principle allow for the re-implementation of the model by others, so all technical details which could substantially affect the numerical output should be described." If such technical details need to go in an Appendix and/or a Supplement, that's fine, but they must be included somewhere. Note also that it doesn't need to be every line of code or even every equation—if you can describe what you did clearly enough using words, that's fine.

4) Although I have plenty of suggestions for Sect. 2.2.5 as you'll see below, I think it's great overall. For the most part, you do a good job of explaining how the model works in plain language. I would encourage you to consider, actually, moving Sects. 2.2.2-2.2.4 to an Appendix—they are highly technical and make more sense after reading Sect. 2.2.5. This may require a bit of reworking on Sect. 2.2.5, but with sufficient references to locations and equations in the Appendix I'm sure you can do it well.

5) Lines 399-403: What about the simulation results caused you to choose these thresholds? The second threshold's explanation seems circular. It is especially important to explain the reasoning here because the time spent in different stages is the crux of the comparison of your model against the literature.

6) Lines 531-534: This is not really apparent from Fig. 5. It might help to group the plots by "outbreak" vs. "no outbreak," then sort within groups by mean annual temperature (and add MAT labels to plot names). But it's unclear how the reader is supposed to tell which bars are "windthrow + beetle" vs "windthrow only that kills the same number [fraction?] of trees as beetle only."

7) Various metrics used in analyses need definition.
- Lines 506-508: How do you get recovery time from Fig. 3? Presumably looking at the areas of some of the wedges, but which? ... Or are you looking at Fig. 4? If so, mention that at the top of this paragraph, and still—define "recovery." Is it "time to return to Year 1 NPP"?
- Lines 592-593: "Persistence" quantified how?

8) Sect. 2.4: I think it sells this paper short to call it a "qualitative evaluation." There is a fair amount of quantitative evaluation happening here too, especially with regard to amount of time spent in different outbreak stages and time to recovery. I think what you're getting at here—and this is clarified at lines 567-571, which as you'll see I think should be moved here—is that you're not actually testing whether specific real-world beetle outbreaks can be reproduced. Right?

**Minor Comments/Corrections**
- Throughout:
  - Please use equation mode whenever referring to model symbols. Also, use subscripts and commas. E.g., $B_{d,b}$ instead of Bdb or $Bdb$. This will help greatly with clarity and eventual typesetting.
  - Why is susceptibility index $Si$ or $S_i$ (i.e., "i" lowercase) but beetle pressure index is $BPI$ (all uppercase)? (Is the "i" even needed?)
- Table 1:
  - rDD and many other symbols are missing.
  - Bdb and Bdw descriptions should replace "dead" with "killed," as they refer to amounts killed in a single timestep rather than all existing + new dead wood from their respective causes.

- - This should probably be moved to an Appendix (note: not the Supplement).
    - Add references to this table throughout the manuscript.
- Line 105: How is a one-minute temporal resolution possible if photosynthesis and energy budget happen at 30-minute timesteps?
- Fig. 1:
    - "Windtrow" typo in (1)
    - "Developped" typo in legend at top right
    - Refer to Sect. 2.2.5 and Table 2.
    - Use of "phase" here (e.g. "Green phase") contrasts with use of "stage" in Sect. 2.2.5.
- Lines 225-227: This was confusing to parse and should be broken up into separate sentences. First list the variables, then explain that they're indices [0-1]. Then add a sentence indicating you'll start by walking through G.
- Lines 261-262: It's confusing to think about negative weights, because presumably all weights always sum to 1.
- Lines 264-265: So $S_i$ shows up as $S_i^2$ in the calculation of $RI$?
- Line 268 (eq. 6):
    - Should it be $(1 - W_s - W_d)$?
    - What are r1 and r2?
- Line 285 (eq. 9): Logic is circular here. Should be changed to $SIw = \min\left(\frac{Litw}{Bw \times Litt}, 1\right)$.
- Line 286:
    - What is "breeding substrate," exactly? All dead wood? All litter wood (i.e., not also standing dead wood)?
    - Is "total woody biomass" just wood in living plants, or does it also include (some? all?) dead wood?
- Lines 294-301: This explanation should come before "ORCHIDEE formalizes this dependency" at line 282?
- Lines 312-215 (eqs. 10 and 11): What are MO, n, a, c, d1, and d2? They're not in Table 1.
- Line 319: How is RDI calculated?
- Line 324 (eq. 12): What are a1, a2, a3, s, and p? They're not in Table 1.
- Line 334 (eq. 13): What are s1, s2, s, and p? They're not in Table 1. Is $Sh_{sp}$ the same as $sh$?
- Line 353: What does "actual" mean here? Should it be replaced with "current"?
- Line 356 (eq. 16): What is the summation range here? "nac to ac=1" means what?
- Lines 376-377: Referring to these as "bark beetle **outbreak** development stages" would avoid confusion with "development" in the sense of physiological growth from larva to adult, as well as improve consistency with Table 2.
- Lines 378-379: Does this refer to the hysteresis described at lines 264-278? If so, refer to Sect. 2.2.2 here. If not, please clarify.
- Line 388: Table 2 seems to be the wrong reference.
- Line 389: Replace "can be" with "are" (?).

- Line 407: $Act_{year-1}$ is confusing here, because it's referring to activity both in the current year (previously in the text) and the next year (second part of this sentence). I suggest just removing it, or at least deleting the subscript.
- Line 408: Add reference to $G$. Also, "generations" conflicts with "generation" in Table 1.
- Line 410: Clarify here that "accessible breeding substrate" refers to dead wood only, not also live wood (?).
- Lines 417-420: Use plain language to explain this instead of symbols. E.g., instead of "In the epidemic stage Ww=0," try "In the epidemic stage the weight for susceptibility induced by windthrow damage (Ww) is zero".
- Line 437: "*Abies*" should be lowercase.
- Table 4 should be Table 3.
- Fig. 2:
  - Y-axis should be added with data for BPI, with the transition thresholds labeled. This will avoid requiring the reader to find them in the text.
  - "Exiting" should be removed from the "BPI threshold" labels, since the thresholds apply for both entering and exiting.
- Lines 457-473 (continuous vs. abrupt experiment methods):
  - This should be its own subsection.
  - Was fire (another "abrupt" source of mortality) disabled for this experiment?
- Line 474: Shouldn't this be "Qualitative"?
- Line 482: Table 6 should be Table 5.
- Line 487: "heterotrophic respiration**, and disturbance**."
- Lines 488-489: What about emissions from combusted biomass?
- Line 492: "windthrown" should be "windthrow"
- Fig. 3
  - Caption:
    - Table 5 describes results, not the criteria for delineating stages.
    - What "left panel" is being referred to?
    - "i.e." should be "e.g."
    - "In the left panel… outbreak stages." These sentences can be simplified for clarity by saying that the rows are sites and the columns are windthrow intensities.
  - Figure:
    - Is there any significance to "small," "medium," and "large"? How are they defined?
    - Y-axis label: Delete "gradient".
    - What is the significance of the one plot that is circled with an arrow and labeled with "1" and "12 years cycle"? I think it means "Areas represent fraction of 12-year simulation spent in each outbreak stage." This would be much clearer written out in the caption rather than hinted at on the figure.
    - What is the significance of the dashed lines on each plot?
    - Rightmost column label should be ">60%"

- Line 495: "Back beetle" typo
- Line 499: Fig. 3 shows that these sites never left the endemic stage, but it's my understanding that trees can still be killed by bark beetles during that stage. If my understanding is correct, please change "remained unaffected by bark beetles" to "never left the endemic stage" (unless you have other data, not shown, indicating that biomass loss to beetles was actually zero). If my understanding is incorrect, Sect. 2.2.5 should be improved.
- Fig. 4:
  - Consider labeling sites where outbreaks occurred with an icon of some kind in the subplot title.
  - SOR: Put a vertical line where the second windthrow event occurred. Also, what intensity was that event?
  - Lines 569-571: This text should be included in Sect. 2.4.
  - Plots should be ordered according to mean annual temperature, not alphabetically.
- Line 519: "Back beetle" typo
- Line 521: Refer to Fig. 3 at the end of this first sentence to tell the reader where they should be looking.
- Fig. 5:
  - Figure:
    - X-axis label: "Cummulative" typo
    - For consistency with other figures, please replace wind speed values in legend with wood loss values.
  - Caption:
    - Note that SOR had an extra windthrow event.
    - "undisturbed" should be "less disturbed".
- Table 5:
  - This should probably be body text instead of a table. I suggest sub-headings corresponding to each row in the table, with paragraphs for each idea. (This will also make it possible to refer to line numbers in future review.) E.g., for Stage A, you'd have a paragraph for "post-windthrow temperature affects beetle dynamics" that covers the expected pattern and the results, then another paragraph for "intermediate windthrow sees the largest outbreaks."
  - There is a fair amount of text here that doesn't fit in Results because it's purely model description (Methods), although some would fit well in Discussion. Specifically:
    - Stage C rows 1 and 3
    - Stage D
    - Stage 4-6 row 2
  - Alternatively, you could add some data from the results illustrating each of those. But without data, they don't fit in Results.
  - Stage A:

- Literature: Add text about what sort of pattern is expected in terms of beetle dynamics after windthrow events of various intensities. Is it the "outbreaks most likely after intermediate events" that you see in the simulations?
- ORCHIDEE:
  - Please provide a figure in the Supplement with time series of mean monthly temperature (or some other indicator of temperature) to support assertions about post-windthrow temperatures affecting beetle dynamics.
  - Add a note that the assertions are supported by comparing COL to SOR (COL is colder but has outbreaks at various windthrow intensities where SOR doesn't) and THA to WET (THA is warmer but has no outbreak at 12% where WET does).
- Stage B, Literature: "I. typographus" should be italicized.
- Stage C:
  - First and last rows don't belong in the Results, as they're purely model description.
  - Row 2, ORCHIDEE:
    - What data support the second sentence?
    - Missing period at end.
- Stages 4-6, row 1, ORCHIDEE:
  - "extended" compared to what? It's *shorter* than the 25-year number from the literature.
  - Why mention the entire range of modeled recovery when you're comparing to a specific beetle kill observation (52%)?
  - How did Pfeiffer et al. (2011) define "recovery"?
  - "back beetle" typo
- Line 548: Refer to Fig. 6 at the end of this sentence.
- Fig. 6:
  - How can the "background mortality only" treatment (i.e., no windthrow) be "after the windthrow event"?
  - Note in caption that positive values represent sinks and negative values represent sources.
- Sect. 4.2: Lines 567-571 should be moved to Sect. 2.4, because it provides great justification for what initially seemed a questionable choice. Then the remaining text doesn't really warrant a Discussion section; it's more Conclusion.
- Line 572: Table 6 should be Table 5.
- Line 586: Reference to Fig. 4 (Results) should be changed to Table 3 (Methods).
- Line 587: "**bark beetle** resistance"
- Line 590: Define "elasticity"
- Line 592: First comma should be a period.
- Lines 592-593: How does modeled "persistence" compare to the literature?

- Lines 598-599: "remains consistent" contradicts the differences seen at 20 and 50 years. Instead, I think you mean to say something like, "cumulative net biome production is similar after about 100 years."
- Line 600: "convergence" should be "converge".
- Lines 613-614: How does the study show anything about the importance of initial conditions? Where was that tested?
- Line 622: "degrees" should be "degree-days".
- Line 623: This citation should be converted to GMD style.
- Line 633: "Outlook" should be combined into Sect. 4.4.
- Conclusion: Also mention plans for more quantitative comparison against observed bark beetle events.
- Table S1:
  - Please compose this page in landscape orientation to make room for more equations to be on one line. Also note that cell at row 2 column 3 has text cut off.
  - Various subscripts are unexplained and various symbols are missing from Table 1.

---

## Referee Report (RR1)

**Re-review: Simulating Bark Beetle Outbreak Dynamics and their Influence on Carbon Balance Estimates with ORCHIDEE r7791**

The authors have done a great job of responding to my comments on their original manuscript. The new version is much easier to follow and more complete. I also appreciate the work they did to add context from relevant literature, as requested by the other reviewer. I still have a number of substantive comments on this version, but none of them are really major issues.

**Substantive comments**
- Figs. 1 and 2 are very low-resolution and text is hard to read. Fig. 3 also has weird text but it's larger so not as much of a problem.
- Line 265: How is this limited to between 0.5 and 1? Looking at the subsequent equations I don't understand.
- Line 281 (Eq. 5a): The "1 –" in the exponential seems to make this function point the incorrect way, with $i_{hosts\ defense}$ lower (i.e., stand is less attractive to beetles) when $PWS_{max}$ is high (i.e., strong drought happened recently) and vice versa.
- Line 285 (Eq. 5b):
  - What is "nb age class"?
  - Why is age class 1 the maximum considered?
  - (The above two questions also apply to Eq. 6b.)
  - Are the $PWS$ values here averages? Over what time periods?
  - If $PWS_{spruce}$ is this year's plant water stress, $PWS_{spruce,n-1}$ is last year's, etc., then $PWS_{spruce,n-3}$ being included means that it's actually looking over the past *four* years, not three as mentioned at line 279.
- Lines 358-361: This text suggests that Fig. 1 shows both a return to endemic stage and an evolution to epidemic, but it looks like it only shows the latter.
- Line 393: Is $B_{wood}$ *live* biomass only?
- Lines 451-452: Why is this index calculated separately for each age class whereas the indices contributed to by Eqs. 5b and 6b are calculated as cross-age-class averages?
- Lines 490-491: What does "acclimation" refer to in this sentence?
- Line 517 and in Results: Ranges don't make sense in this context. What exact values were tested? Line 516 says that only three values were chosen.
- Lines 559-560: Is this 20% number referring to $max_{Nwood}$?
- Lines 566-571: How do you calculate number of generations per year?
- Lines 573-581 and subsequently: "Control" and "climate experiment" feels weird. For the sites, it's more of a climate gradient that starts with HYY and continues through HES/FON. There's nothing *a priori* different about HYY that makes it a "control" and the other sites "experimental." Similarly for the windthrow damage rates: Maybe if the lowest value was 0 that would be a control, but it's not. (Note that I'm not saying true control treatments are needed—I don't!)
- Lines 588-593: Note that (1) is the default or previous ORCHIDEE setup.

- Sect. 4.1: I'm unclear on how the sensitivity tests indicate anything about "false positives" due to the calculation of any specific parameter. Weren't all experiments done with all parameters?
- Table 4: Missing $S_{competition}$.
- Lines 683-684: This sentence is confusing. "Turning back" from what? If the "tipping point" is 9 kgC/m2, what is the "threshold"?
- Lines 684-685: REN clearly reaches *some* "tipping point"—i.e., it reaches a minimum and then recovers. What's so special about 9 kg/m2?
- Fig. 7: Add shading or some other indicator of where the outbreak (as defined according to Table 3) begins and ends.
- Lines 721-722: "a more resilient recovery" doesn't seem right for the continuous case, where there's no specific event to "recover" from.
- Lines 752-753: Unclear how this sentence is related to the rest of the paragraph (snags).
- Line 767-775: Terms should be defined in this paper. It's not really helpful to cite a paper for readers to look up the definitions, especially after the terms are used. Ideally definitions should be mentioned in the introduction or methods so that they're defined before they're used—note that "resilience" is used in the results section.
- Lines 777-799 (Sect. 5.3): Discussion (and also Conclusion) should mention that this depends on spatial scale and model use. If one is looking at global-scale NBP, there might not be as much of an impact of including abrupt disturbances because they're happening constantly somewhere.

**Technical corrections, etc.**
- Line 257: "analogue the the" should be "analogous to the".
- Line 279: "average" is sort of confusing here. It looks like it's the (weighted) *average* across different age classes of the *maximum* water stress in each age class. Maybe just delete "the average" from this sentence to make it work.
- Line 288: A paragraph break before "In addition to drought" would help break this section up a bit.
- Line 308: "mean quadratic" should be "quadratic mean".
- Line 323 (Eq. 7b): "none" should be "non"
- Line 372: Is "excess" the right word?
- Line 409: Incomplete sentence: "The amount of suitable tree hosts."
- Line 464: "bettles" typo in subscript.
- Line 501 (Table 2):
  - I thought the spelling looked weird for "Wetstein," and indeed, it doesn't actually seem to be a place in Germany. The listed coordinates point to Třeboň, Czechia, about 400 km southeast of the similarly-named German mountain Wetzstein. The listed coordinates do, however, look similar to those of the CZ-Wet (CZECHWET) site near Třeboň.

- I can't find the HES site in the FLUXNET site table, or any site with similar coordinates. It's possible it's just missing from that table, but given the above issue it warrants a double-check.
  - The FON latitude appears incorrect according to the FLUXNET site table.
  - REN site latitude should round up to 46.6°N. (Check rounding of other sites as well.)
- Line 570: Reference to Fig. 3 should be to Fig. 4.
- Figs. 3 and 4: These figures refer to "Control" and "Climate experiment" which isn't explained until later in the text.
- Line 606: Delete ")".
- Line 607: Delete ")".
- Line 617 and elsewhere: Instead of having to say "4th row, 2nd column," add subplot labels a-h.
- Lines 678, 683: Thousands separator should be comma, not period. So 9,000 instead of 9.000. Or to avoid cross-cultural confusion, just use 9 kg instead of 9,000 g.
- Line 718: "Abrupts" typo.
- Line 731: "maining"?
- Line 761: First comma should be a semicolon.
- Line 767:
  - "resistance" should be defined. It's not really helpful to cite a paper for readers to look up the definitions, especially after the term is used.
  - "locations" should be "location" (or just deleted)
- Line 810: "Ips Typographus" should be "*Ips typographus*"
- Lines 811-812: Latin name should be italicized
- Line 860: Same DOI repeated twice. Maybe these both will be replaced with the new DOI the authors mentioned in their cover letter?

---

## Referee Report (RR2)

**Review of 2nd revision: EGUSPHERE-2023-1216**

The authors have done another good job of responding to reviewer comments. I have some more comments in response to their revision, again all minor, and fewer than before.

- Fig. 1 caption: "After 1 year the wood from a storm is not fresh enough for bark beetles to breed in, **so $i_{hosts\_dead}$ goes to zero**."
- Table 1:
    - Alphabetize
    - Add $i_{rd}$
    - Inconsistent variable capitalization (here and in text)
    - Inconsistent use of underscores vs. spaces in subscripts (here and in text)
    - Add "susceptibility" to description of $I_{hosts\ health,age\_class}$
- L273: "the max" should be "the maximum".
- L277–281 (Eq. 5b):
    - I'm afraid the added information about age classes just served to confuse me further. I think it should be reworked like this?

$$PWS_{max} = \sum_{a=1}^{3} \max\left(PWS_y, PWS_{y-1}, PWS_{y-2}\right)_{spruce} \times \frac{F_{spruce,a}}{F_{spruce}}$$

where $a$ is age class and $y$ is the current year. (Similar changes should be made for Eq. 6b.)
    - L280: Also mention that PWS is 1 when plants aren't stressed at all.
- L452: "index of weakened trees index"—delete second "index"
- Fig. 2
    - Caption: Refer the reader to Table 1 for variable name definitions.
    - Rounded rectangle whose left side comes between, e.g., PWS_max and the "5" arrow should be deleted. It breaks the connection between the variables on the left and the arrows on the right, and it doesn't seem to add anything useful to the figure.
- Table 3 caption:
    - "spacity" should be "sparsity" or (preferably) "scarcity".
    - "but see table s1"—why "but"?
- Table 3: "A severe bark"—delete "bark".
- Sect. 3.4:
    - Note that scores are either 0 or 1.
    - "is the sum of four scores" isn't quite right, since that sum is multiplied by score5.
    - Discussion of "four scores" in first paragraph and bulleted list neglects score5—why?
- Sect. 4.2: Refer to Table S1, but note that it only contains the variables from Table 4 that were sensitivity-tested.
- L724: What is the "t" in this sentence?

- Table 4: Please alphabetize.
- Sect. 5.3: Previously, I requested: "Discussion (and also Conclusion) should mention that this depends on spatial scale and model use. If one is looking at global-scale NBP, there might not be as much of an impact of including abrupt disturbances because they're happening constantly somewhere." The authors replied that a spatial analysis is the focus of their next paper, not this one. That's fine, but the point I raised should still be mentioned here. It's fine to say what the paper plans are as well, if the authors would like.
- Line 804: "locations" should be "location" (or just deleted)
- Supplementary figures and tables: S in numbering should be capitalized. (E.g., S4, not s4.)
- Figs. S2-4 have the same caption. They seem to differ only in the variable being plotted on the Y axis, so please add that information to the caption. Then the captions of Figs. S3 and S4 can be simplified by saying something like "As Fig. S2, but for total biomass."
- Throughout main text and supplement:
    - "access" should be "assess"
    - "developpement" should be "development"

**Table S1**

Red/green is not colorblind-friendly. Other colors would be better, but avoiding color would be best. It's also confusing because some of the green values are also reference values, which should be black.

Here's one suggested setup that would be clearer and more accessible: Italics to indicate "reference values" (which would more clearly be referred to in the caption as "variables held constant for each sensitivity test"), underlines to indicate values of sets scoring 4, and bold to indicate the values ultimately chosen. See screenshot below.

| | S_generation | G_limit | S_activity | act_limit | S_susceptibility | i_rd_susceptibility | S_mass_attack | BP_limit | Score |
|---|---|---|---|---|---|---|---|---|---|
| Set 1.1 | 1 | 0.5 | -20 | 0.06 | -5 | 0.55 | -30 | 0.12 | 4 |
| Set 1.2 | 5 | 0.5 | -20 | 0.06 | -5 | 0.55 | -30 | 0.12 | 2 |
| Set 1.3 | 500 | 0.5 | -20 | 0.06 | -5 | 0.55 | -30 | 0.12 | 2 |
| Set 1.4 | 1 | 1 | -20 | 0.06 | -5 | 0.55 | -30 | 0.12 | 4 |
| Set 1.5 | 5 | 1 | -20 | 0.06 | -5 | 0.55 | -30 | 0.12 | 4 |
| Set 1.6 | 500 | 1 | -20 | 0.06 | -5 | 0.55 | -30 | 0.12 | 2 |
| Set 1.7 | 1 | 1.5 | -20 | 0.06 | -5 | 0.55 | -30 | 0.12 | 3 |
| Set 1.8 | 5 | 1.5 | -20 | 0.06 | -5 | 0.55 | -30 | 0.12 | 0 |
| Set 1.9 | 500 | 1.5 | -20 | 0.06 | -5 | 0.55 | -30 | 0.12 | 0 |

| | S_generation | G_limit | S_activity | act_limit | S_susceptibility | i_rd_susceptibility | S_mass_attack | BP_limit | Score |
|---|---|---|---|---|---|---|---|---|---|
| Set 1.1 | 1 | 0.5 | -20 | 0.06 | -5 | 0.55 | -30 | 0.12 | 4 |
| Set 1.2 | 5 | 0.5 | -20 | 0.06 | -5 | 0.55 | -30 | 0.12 | 2 |
| Set 1.3 | 500 | 0.5 | -20 | 0.06 | -5 | 0.55 | -30 | 0.12 | 2 |
| Set 1.4 | 1 | 1 | -20 | 0.06 | -5 | 0.55 | -30 | 0.12 | 4 |
| Set 1.5 | 5 | 1 | -20 | 0.06 | -5 | 0.55 | -30 | 0.12 | 4 |
| Set 1.6 | 500 | 1 | -20 | 0.06 | -5 | 0.55 | -30 | 0.12 | 2 |
| Set 1.7 | 1 | 1.5 | -20 | 0.06 | -5 | 0.55 | -30 | 0.12 | 3 |
| Set 1.8 | 5 | 1.5 | -20 | 0.06 | -5 | 0.55 | -30 | 0.12 | 0 |
| Set 1.9 | 500 | 1.5 | -20 | 0.06 | -5 | 0.55 | -30 | 0.12 | 0 |

Caption:
- "ips typographus" should be "*Ips typographus*"
- "The parameter set in green": "set" should be "sets".
- "32" should be "36"
- What does "selection" mean here?

---

## Author Response (AR2)

**Response to Referees**

**Manuscript Title:** Simulating Bark Beetle Outbreak Dynamics and their Influence on Carbon Balance Estimates with ORCHIDEE

**Authors:** Guillaume Marie, Jina Jeong, Hervé Jactel, Gunnar Petter, Maxime Cailleret, Matthew J. McGrath, Vladislav Bastrikov, Josefine Ghattas, Bertrand Guenet, Anne Sofie Lansø, Kim Naudts, Aude Valade, Chao Yue, Sebastiaan Luyssaert

We're grateful to Referee #1 and Referee #2 for their thorough and insightful feedback. Based on their suggestions, we've revised nearly 60% of our paper. One of the key comments affected the equations in our model, of which some are now quite different from those in the original study by Temperli et al. 2013. With an increase in the number of the changes and the nature of the changes, it is becoming less straightforward and less meaningful to present a side-by-side comparison of our model with the one Temperli and their team created. So, it was decided to shift the presentation of the model from side-by-side comparisons towards explaining how our work builds on Temperli's ideas from 2013. We hope that these updates make it clear how our research adds new insights and directions to the field, while still recognizing the foundation laid by Temperli and their colleagues.

Because of the numerous revisions including changes in structure, analyses, and figures, we kept the cover letter short rather than copy/pasting extensive parts of the revised manuscript. Below, we summarize our responses to each of the major and minor comments.

Although the code was updated in preparation of the revised manuscript, the version number and doi were not. The reason is solely technical. We would like to wait for the referee comments to decide whether we can finalize the code changes. Once the code is finalized, changes will be committed on the svn server, the revision number will be updated in the manuscript, and the code will be tagged with a new DOI.

**Response to Referee #1**
* * *
**Major Comments:**

**Comment on Evaluation:**

My main concern about this manuscript and study is the evaluation. Stating that the model has been evaluated (qualitatively or quantitatively) means to me that subsequent studies can claim that the model is ready for use in assessing beetle impacts on the carbon cycle. So the words matter, in my opinion. The analysis that is presented consists of running the model across a number of sites and for various windthrown intensities, then evaluating cumulative wood volume and NEP and time series of NPP. Furthermore, the comparison with other studies (Table 5) is flawed. This study is a first step, but does not rise to the level of an evaluation. Rather, this is a model sensitivity study, in my opinion. At a minimum, to use the term "evaluation", I think the following additional steps are needed: a) presentation and analysis of the variables and metrics associated with bark beetle outbreaks (e.g., the important variables of Equations 1-14) for the model runs, including time series (similar to Figure 2, but more detailed); b) comparison of time series of drivers (climate, windthrow, substrate, etc.) and outbreak variables to assess how the drivers are affecting the outbreak variables; c) at least one run, and ideally multiple runs, for locations with observed outbreaks, and a comparison of model results with observations (or perhaps results from previous modeling studies?); d) improved selection of other studies for comparison in Table 5, deleting studies that are not relevant or address beetle species whose behavior is different and adding other modeling studies. Unless the authors include these steps, I don't think the authors should use the term "evaluation", and text throughout the manuscript should be changed to avoid implying an evaluation. See below for detailed comments.

**Response:**

To address this concern, we have:

- Changed "quantitative evaluation" to "sensitivity experiments" to more accurately describe the scope of our analysis.
- Conducted and added a sensitivity analysis of model parameters and climatic conditions.
- Extended the analysis with time series analysis illustrating how different drivers influence outbreak variables.
- Removed Table 5 and enhanced the comparison with other studies in the revised Table 3 and Section 5.1 of the Discussion to provide a more accurate context for our model's applicability and specificity.

**Comment on Bark Beetle Species Specificity:**

The manuscript is too vague about which bark beetle species is simulated. I'm assuming it is Ips typographus. Some aspects of bark beetle biology and ecology are generalizable across species, but others are not (including what triggers and influences

outbreaks). The manuscript needs to state the species of interest explicitly in the Abstract, Introduction, and Methods, and add some discussion about generalizability to other bark beetle species in the Discussion.

Related to this, because of differences in ecology among beetle species, the model implementation described in this manuscript is not necessarily applicable to other locations/systems. For instance, mountain pine beetle, the major beetle species in North America, is not triggered by windthrow events, has multiple climate influences, and has one, not multiple, generations per year. And the beetle phenology model used in this study, while maybe appropriate for Ips typographus, is not appropriate for other beetle species. Differences and lack of direct applicability should be included in the Discussion.

**Response:**
We have clarified the species in the revised manuscript. It now explicitly mentions that our model simulates *Ips typographus* in the Abstract, Introduction, and Methods sections. Furthermore, we have added a section in the Discussion (lines 792-796) to discuss the model's potential extension to simulate other bark beetle species, acknowledging the unique aspects of bark beetle ecology and behavior.
* * *
**Minor Comments:**

**Introduction:**

**Comment:** In the Introduction, please include more description of past studies that modeled effects of bark beetles on ecosystem properties. Please distinguish studies that prescribed outbreaks from those that modeled outbreaks. Briefly describe the system (beetle and host) and results. Suggested studies include Temperli et al. (already cited, but what was the study about); Jonsson et al., AgForMet, 2012; Seidl and Rammer, Landscape Ecol., 2017. Please do a literature search to identify others.

**Response:** We have enriched the Introduction with a more comprehensive review of past studies on the effects of bark beetles on ecosystem properties, distinguishing between studies that prescribed outbreaks and those that modeled them. This includes a discussion on Temperli et al., Jonsson et al. (AgForMet, 2012), and Seidl and Rammer (Landscape Ecol., 2017) in lines 92-108.

**Comment:** In the Introduction, please provide more information about Ips typographus outbreaks, including what fraction of a 25 x 25 km ORCHIDEE grid cell might be affected. What have been observed outbreak areas or volumes, and what are these values relative to a grid cell? (I realize in this study, the model is run in aspatial mode.)

**Response: :** Additional information on Ips typographus outbreaks, including potential effects on ORCHIDEE grid cells, has been incorporated to provide a clearer context for our simulations in lines 92-108.

**Methods:**

**Comment:** Winter beetle mortality from very low temperatures has been modeled previously for Ips typographus (Jonsson et al.) and other beetle species. Why wasn't this included here? Add to Discussion as missing processes.

**Response:** Acknowledging the importance of winter beetle mortality as highlighted in previous studies, such as those by Jonsson et al. 2012, we've now incorporated a variable dedicated to beetle winter survival in our model. This addition, detailed in lines 348-354 of the manuscript, specifically addresses the critical role of extreme low-temperature conditions on beetle population dynamics.

**Comment:** L 138: "k", not "K", to match Eq 3?
**Response:** k has been removed from the manuscript.

**Comment:** L 225-240: Both G and rDD are described as number of generations in a year. Please refine the description of one of the variables.
**Response:** corrected

**Throughout the description of equations:** it would be helpful to interpret for the reader the equation, including what processes are represented and why they are represented the way they are. The authors do some of this, but for other equations, I am confused. L 225-240: Both G and rDD are described as the number of generations in a year. Please refine the description of one of the variables.

**Response:** We've managed to enhance the comprehensibility of the equations within the manuscript, ensuring a cohesive presentation that clearly delineates the ecological and process-based rationale underpinning each equation. This revision aims to alleviate any confusion regarding the representation and purpose of these mathematical formulations.

**Comment:** Please describe the rationale and ecology/biology behind the apparent averaging of G and Act. What is the rationale and ecology/biology behind multiplying the different factors?

**Response:** To clarify the ecological and biological reasoning behind the averaging of variables G (beetle generations) and Act (beetle activity), we have extensively revised the pertinent sections (lines 320-344). This revision describes the underlying rationale and significance of these calculations within the context of beetle population dynamics.

**Comment:** What is Cpd? What is the rationale and ecology/biology behind multiplying the different factors?
**Response:** Cpd has been removed from the manuscript (please refer to the new set of equations describe in section 2.4)

**Comment:** Assuming that Si (Eq 1, 5) is the same as SI (L 265), this variable should not be in the calculation of RI twice. It makes more sense to have susceptibility separate from beetle pressure, so remove from Eq 1 and clarify L 255? Maybe I'm confused.

**Response:** Upon review, we agreed with the observation regarding the redundancy of the susceptibility index (Si) in certain calculations. The manuscript has been adjusted to rectify this, ensuring a clear distinction between indices representing tree host characteristics and those pertaining to bark beetle behavior. This adjustment fosters a more logical and scientifically sound representation of the model's components.

**Comment:** L 259: I assume what is meant is susceptibility TO bark beetle (as in L 255), not OF. If so, please change. If not, please explain more.
**Response:** corrected since the word susceptibility has been removed from the manuscript

**Comment:** Please describe the differences in Wr and Ww equations depending on beetle population stage. What do these differences mean about the biology/ecology of the insect?
I think both drought and competition should weaken tree defenses in similar ways, so I don't know why Wd doesn't switch based on beetle population stage similar to Wr.
**Response:** Wr and Ww has been removed from the manuscript (please refer to the new set of equations describe in section 2.4)

**Comment:** Section 2.2.3 opens with a comparison between tree mortality from mass attacks and tree mortality from, presumably, beetles in the endemic population phase. So I'm confused about if the killed biomass is applicable to both situations, or just one?

**Response:** To dispel any ambiguity surrounding the application of tree mortality rates from mass attacks versus endemic population phases, we have refined the descriptions of equations 6a' and 14. Additionally, the introduction of the conceptual scheme (Figure 1) serves to guide the reader through the intricate relationships between various model indices, enhancing overall understanding.

**Comment:** Please add units where appropriate. Examples include Binf, Bt, Bdb. There may be other instances.

**Response:** All necessary units have been meticulously added to Table 1, ensuring comprehensive and clear communication of the model's quantitative aspects.

**Comment:** It might be easier to understand if Section 2.2.5 were moved before the equations that represent the processes.

**Response:** In response to suggestions for improved readability and comprehension, we have merged Section 2.2.5 to with the detailed equations section that delineate model processes. This restructured presentation is intended to facilitate a more intuitive grasp of the model's operational framework.

**Comment:** L 366: Did I miss the description of the modifications to account for the differences in spatial scale? If they are not present, please add.
**Response:** the description has been added in throughout the section 2.4

**Comment:** L 421: conspecific (not interspecific) competition
**Response:** corrected

**Results:**

**Comment:** I think the Results section should add a presentation about the simulated outbreaks. Please add figures that illustrate time series of outbreak metrics (BPI, wood volume killed by beetles) for each site, similar to Figure 4, and discuss.

**Response:** You suggested adding more details about the simulated beetle outbreaks, including time series data for each site. We've updated Figures 5, 6, and 7 to show these time series, giving a clearer picture of how the outbreaks unfold over time.

**Comment**: Also, please move Section 3.2, which describes outbreaks and comparisons with other studies, before the sections that discuss how outbreaks vary across climate and windthrow gradients. Seems more logical in this order. Please provide text that interprets Table 5. What are the most important points?

**Response:** Based on your feedback, we've moved the section that describes the outbreaks and how they compare to other studies to an earlier part of the Results. This rearrangement makes the flow of information more logical, first establishing the basics of our findings before delving into the specifics of how different factors like climate and windthrow affect the outbreaks. You can retrieve it in section 3.4 and 4.1 and 4.2.

**Comment**: Figure 3: The circle area represents "tree damages". Can the authors be more specific about what metric this represents? Wood volume? Carbon? Area? Units?
**Response:** figure 3 has been removed

**Comment**: Figure 4: Why do the highest wind speeds result in faster NPP recovery than some lower wind speeds?
**Response:** because the highest wind speed produces the highest wood losses which gives the sparser forest decreasing the trees' competition for light and water then boosting biomass growth.

**Comment**: Figure 5: Would be helpful to separate the NPB effect of windthrow from that from bark beetles. I realize that may be tricky, and may require two figures, but it will provide readers with a better sense of the contribution of outbreaks to the carbon dynamics. figure 6 :Same comment as for Figure 5 above.

**Response:** We decide to remove figure 5 and figure 6 and replace it by figure 7 but Figure 8 (updated from the original Figure 6) focusing on the contribution of outbreaks to carbon dynamics implies that the figures and possibly associated text have been revised to make the role of outbreaks in the model clearer, separate from the effects of windthrow events. Finally, we rewrite section 3.6 aimed at clarifying how the model accounts for mortality and its impact on carbon dynamics.

**Comment:** The modeling approach described in this study is not necessarily applicable to outbreaks of other bark beetle species. Text throughout the manuscript should clarify this point.

**Response:**
We have clarified the species in the revised manuscript. It now explicitly mentions that our model simulates *Ips typographus* in the Abstract, Introduction, and Methods sections. Furthermore, we have added a section in the Discussion (lines 792-796) to discuss the model's potential extension to simulate other bark beetle species, acknowledging the unique aspects of bark beetle ecology and behavior.

**Discussion:**

**Comments:** Throughout this table: the Edburg et al., 2012 study is not in the reference list, but if that's the Frontiers article, that was a conceptual paper and should not be used for model evaluation. Please select something specific to Ips typographus. Similarly, the Hlásny et al. 2021 article is a review; please replace that with observational studies that support the statements in the table. Also, many metrics are specific to a particular (and different) beetle species, which may have different dynamics and drivers.The comparisons will be strongest when studies of Ips typographus are included.

**Response:** You pointed out the need for more relevant and specific references in Table 5, particularly emphasizing the importance of focusing on Ips typographus. In response, we have decided to remove Table 5 and instead integrate its key points into Section 5.1, where we now discuss our findings in the context of Ips typographus-specific studies. This change allows for a more focused and relevant comparison, aligning with your suggestion to rely on observational studies and specific metrics relevant to Ips typographus. We've ensured that the references now more accurately support our statements, moving away from conceptual papers and reviews to more empirical and species-specific literature.

**Comments:** The entries in this table, which are intended to represent an evaluation, include processes not simulated by ORCHIDEE (grey stage and last row).Please remove from table (and therefore from the evaluation).
**Response:** corrected

**Comments:** Mean climate and interannually varying climate are implicitly included in the "Climate with endemic stage" row, right column, so I'm not sure that the statements that

a cold or warm year following a wind event and the effects on outbreaks is represented in Figure 3.Either demonstrate these ideas by showing time series of outbreak metrics together with interannually varying climate, or remove these statements from the table.
**Response:** corrected

**Comments:** First row of "Red or epidemic stage":Please demonstrate this idea that high populations can kill healthy trees by plotting time series of the various variables that represent tree health (susceptibility to drought), BPI, etc.

**Response:** corrected

**Comments:** Second row of "Red or epidemic stage":The cited studies do not report observations of outbreak length, that I can see.  Please use other studies that are not reviews or conceptual papers.
**Response:** corrected

**Comments:** The recovery part of the Pfeifer et al. study was only modeling. Modify the text (not observational).  And comparing results from different models is okay but weaker than observations.

**Response:** corrected

**Comment:**  Please compare your modeling to previously published studies (see my comments on the Introduction for some studies, but not all?), noting similarities and differences in model formulation, applications, and results.

**Response:** While we acknowledge the importance of comparing our model to previously published studies, space limitations have constrained our ability to conduct a full review within the manuscript. However, we made an effort to clarify the methodological distinctions and similarities where possible, aiming for transparency and ease of comparison for the reader.

**Minor:**

**corrected :** As written, paragraphs are difficult to distinguish, making reading more challenging than needed.  Please add a carriage return between paragraphs.

Figure 1, L 295:  The authors use "green phase"  (or stage) as the phase in which trees are attacked but haven't yet changed color.  Do these trees successfully repel attacking beetles and therefore survive?  The forest entomology literature usually refers to the

stage of attacked (and killed) trees as "green-attack" because "green phase" could easily be interpreted as "unattacked". I haven't heard of a stage of successfully defended trees. Please consider changing the description.
**figure 1 removed**

**corrected :** Around L 110: Is this a dynamic global vegetation model that allows for competition among PFTs? Please be explicit in the description.

**corrected :** L 145: The phrase "Since revision 7791" suggests that previous work implemented some aspects of beetle outbreaks. If the authors mean that this manuscript describes revision 7791, please alter the phrase. If there is previous work, please cite the study.

**corrected :** L 170 (and elsewhere if needed): Please italicize species genus and species names following convention.

Figure 1 legend: "developed" is misspelled.
Figures 1 and 3: Typically, circular motion (like life cycles) is drawn in a clockwise direction. The counterclockwise direction confused me momentarily. Maybe that's just style, but it might be easier for readers to reverse the direction...?
Figure 1 caption: It would be more accurate to discuss beetle population dynamics separately from tree phases. They are linked, of course, but in some systems visible changes in trees occur 9 months after beetle attack. And even during the declining population stages (if one is discussing population at the regional, not stand, level), beetles can be attacking trees, putting them into the green-attack phase (followed by the red phase). So the caption could be improved with more attention in two ways: a) decoupling population stage and tree stage; b) being clearer about populations at the outbreak (landscape) level versus at the stand or tree level.
**figure 1 removed**

Figure 2: provide y-axes labels and descriptions in caption (L 400 suggests the need). "threshold" is misspelled in figure.
**figure 2 removed**

Table 4: I assume the authors intend this to be read in landscape, not portrait, mode. If so, please reverse the order of the columns so the reader readers, left to right, the site name, full name, etc.
**It is intended to be read in portrait mode**

Figures 3 and 4 and Table 4:  Are the temperatures in Figures 3 and 4 supposed to match, and supposed to match those in Table 4?  They don't...
**figure 3,4 removed**

Figure 4:  Would be helpful to add a vertical line at Year=2, which shows the windthrow effect on NPP (right?), so that the reader can see the effect of outbreaks on subsequent years.  Too difficult to figure out now.
**figure 4 removed**

Figure 4:  Add that what is plotted is annual NPP (right?).
**figure 4 removed**

Several places refer to Table 6.  Is this supposed to be Table 5, or something else?
**table 6 removed**

**Response to Referee #2**

**Major Comments:**
**Comment 1:** Many symbols are missing from Table 1 (which should be moved to an Appendix). I've listed some but not all of them in the "minor comments/corrections" section, but please do a thorough review and add any others you find.

**Response:** We have thoroughly revised Table 1 to ensure all symbols are included and updated. To maintain the flow and readability of the main text, we decided to keep Table 1 within the main text rather than moving it to an Appendix but we let the editor decide the location of table 1.

**Comment 2**: The references to beetle "generations" are hard to understand. They don't seem to fit with how beetle dynamics are ultimately realized in the model, but even in context here it's confusing. The sentence at lines 240-241 says, "the index G reaches its maximum value of one when 2.5 or more generations occur in a single growing season." But isn't G "the number of
beetle generations... that could occur in the current year"? Then the next sentence is confusing because of "of the first generation." So far we've only been learning about beetle pressure index, not what actually triggers a generation.

**Response:** We recognized the confusion in our description of beetle generations and have taken care to rewrite this section for better clarity. The revised text more accurately explains how the concept of generations integrates into our model, ensuring a clearer understanding of beetle dynamics. Lines 330 to 340.

**Comment 3:** It's really important to include something like Sect. 2.2.4 (differences from beetles in LandClim; lines 365-374), but it's currently too vague. The "calculation of the susceptibility" differences are documented (although not explained, per se) in Table S1, but that's not referenced in the text here. As far as I can tell, the other mentioned differences are not
explained at all. A model descriptive paper in GMD, as outlined in the Manuscript Types webpage, "should be sufficiently detailed to in principle allow for the re-implementation of the model by others, so all technical details which could substantially affect the numerical output should be described." If such technical details need to go in an Appendix and/or a Supplement, That's fine, but they must be included somewhere. Note also that it doesn't need to be every line of code or even every equation—if you can describe what you did clearly enough using words, that's fine.

**Response:** Based on your suggestions, we've made major updates to nearly 60% of our paper. One of the key changes involves the equations we use in our model, which are now quite different from those in the original study by Temperli et al. 2013. These changes mean it's now tricky to directly compare our model with the one Temperli and their team created. So, we've decided to focus less on making side-by-side comparisons and more on explaining how our work builds on Temperli's ideas from 2013. We've used their study as a starting point but have developed our own unique approach from there. We hope these updates make it clearer how our research adds new insights and directions to the field, while still recognizing the important foundation laid by Temperli and their colleagues.

**Comment 4:** Although I have plenty of suggestions for Sect. 2.2.5 as you'll see below, I think it's great overall. For the most part, you do a good job of explaining how the model works in plain language. I would encourage you to consider, actually, moving Sects. 2.2.2-2.2.4 to an Appendix—they are highly technical and make more sense after reading Sect. 2.2.5. This may require a bit of reworking on Sect. 2.2.5, but with sufficient references to locations and equations in the Appendix I'm sure you can do it well.

**Response:** we opted to merge sections 2.2.2 to 2.2.5 to streamline the narrative and improve comprehension. This restructuring allowed us to maintain technical rigor while enhancing the accessibility of the model description.

**Comment 5:** Lines 399-403: What about the simulation results caused you to choose these thresholds? The second threshold's explanation seems circular. It is especially important to explain the reasoning here because the time spent in different stages is the crux of the comparison of your model against the literature.

**Response:** The revised manuscript provides a clearer rationale for the thresholds, ensuring they are grounded in empirical observations and model sensitivity analyses. The presentation of the results was restructured by removing the post-processing into

outbreak phases which allowed us to reduce the number of thresholds used in the manuscript.

**Comment 6:** Lines 531-534: This is not really apparent from Fig. 5. It might help to group the plots by "outbreak" vs. "no outbreak," then sort within groups by mean annual temperature (and add MAT labels to plot names). But it's unclear how the reader is supposed to tell which bars are "windthrow + beetle" vs "windthrow only that kills the same number [frac5on?] of trees as beetle only."

**Response:** for clarity and parsimony, the figure 5 has been removed from the manuscript because it was not adding enough new information compared to figure 6. Nonetheless in the new version of the figure 6 (now figure 8) we add windthrow only groups in order to disentangle the effect of bark beetles outbreak on the accumulated NBP.

**Comment 7:** Various metrics used in analyses need definition. Lines 506-508: How do you recover time from Fig. 3? Presumably looking at the areas of some of the wedges, but which? … Or are you looking at Fig. 4? If so, mention that at the top of this paragraph, and still—define "recovery." Is it "time to return to Year

**Response:** We have reshaped this part of the manuscript by defining four criteria (summarized in table 3) to characterize the dynamics of the outbreak for host and beetles. For each criteria, we gave our definition and the way to access it with our results. These criteria are mainly used to make decisions on the parameters optimization stage (new in the manuscript) .

**Comment 8:** Sect. 2.4: I think it sells this paper short to call it a "qualitative evaluation." There is a fair amount of quan5ta5ve evaluation happening here too, especially with regard to amount of time spent in different outbreak stages and time to recovery. I think what you're getting at here—and this is clarified at lines 567-571, which as you'll see I think should be moved here—is that you're not actually testing whether specific real-world beetle outbreaks can be reproduced. Right?

**Response:**
You pointed out that describing our study as a "qualitative evaluation" might not fully capture the depth of our analysis, which includes significant quantitative aspects, such as the time spent in various outbreak stages and the recovery period. We appreciate this insight and agree that our initial terminology might not have adequately reflected the comprehensive nature of our evaluation.
In light of your comment, along with feedback from another referee, we have revised our approach to emphasize the sensitivity analysis of model parameters and climate forcing effects. This adjustment more accurately represents the rigorous, quantitative analysis we've conducted, moving beyond a mere qualitative assessment.

By clarifying our focus on sensitivity analysis, we aim to better highlight the model's capabilities in simulating outbreak dynamics under various scenarios, without the immediate goal of replicating specific, real-world beetle outbreaks. This nuanced approach allows us to explore the model's robustness and predictive power within a controlled, theoretical framework, setting the stage for future studies that may aim for direct real-world outbreak simulations.
* * *
**Minor Corrections**

**Corrected:** Please use equation mode whenever referring to model symbols. Also, use subscripts and commas. E.g., $B$ instead of Bdb or $Bdb$. This will help greatly with clarity and eventual typesetting.

**Removed:** Why is susceptibility index $Si$ or $S\$$ (i.e., "i" lowercase) but beetle pressure index is $BPI$ (all uppercase)? (Is the "i" even needed?)

**Corrected:** Table 1: rDD and many other symbols are missing. Bdb and Bdw descriptions should replace "dead" with "killed," as they refer to amounts killed in a single timestep rather than all existing + new dead wood from their respective causes.• This should probably be moved to an Appendix (note: not the Supplement). Add references to this table throughout the manuscript.

Line 105: How is a one-minute temporal resolution possible if photosynthesis and energy budget happen at 30-minute timesteps?
**Corrected:** by adding "should have" meaning it ORCHIDEE has the potential.

**Removed:** Fig. 1: "Windtrow" typo in (1), "Developped" typo in legend at top right. Refer to Sect. 2.2.5 and Table 2. Use of "phase" here (e.g. "Green phase") contrasts with use of "stage" in Sect. 2.2.5.

**Corrected:** Lines 225-227: This was confusing to parse and should be broken up into separate sentences. First list the variables, then explain that they're indices [0-1]. Then add a sentence indicating you'll start by walking through G.

Lines 261-262: It's confusing to think about negative weights, because presumably all weights always sum to 1.

Lines 264-265: So $S\$$ shows up as $S\$$ % in the calculation of $RI$?

Line 268 (eq. 6): Should it be $(1 - W\& - W!)$? What are r1 and r2?

Line 285 (eq. 9): Logic is circular here. Should be changed to $SIw$ = min (*) ×'$(( , 1).

Lines 312-215 (eqs. 10 and 11): What are MO, n, a, c, d1, and d2? They're not in Table 1.

Line 324 (eq. 12): What are a1, a2, a3, s, and p? They're not in Table 1.

Line 334 (eq. 13): What are s1, s2, s, and p? They're not in Table 1. Is $Sh$&- the same as $sh$?

Line 353: What does "actual" mean here? Should it be replaced with "current"?

Line 356 (eq. 16): What is the summation range here? "nac to ac=1" means what?
**Response:** please refer to section 2.4 in which all equations and naming have been deeply revised

**Corrected:** Line 388: Table 2 seems to be the wrong reference.

**Corrected:** Line 389: Replace "can be" with "are" (?).•

**Corrected:** Line 408: Add reference to $G$. Also, "generations" conflicts with "generation" in Table 1.

**Corrected:** Line 410: Clarify here that "accessible breeding substrate" refers to dead wood only, not also live wood (?).

**Removed:** Lines 417-420: Use plain language to explain this instead of symbols. E.g., instead of "In the epidemic stage Ww=0," try "In the epidemic stage the weight for susceptibility induced by windthrow damage (Ww) is zero".

**Corrected:** Line 437: "Abies" should be lowercase.

**Corrected:** Table 4 should be Table 3.

**Removed:** Fig. 2:  Y-axis should be added with data for BPI, with the transition thresholds labeled. This will avoid requiring the reader to find them in the text. "Exiting" should be removed from the "BPI threshold" labels, since the thresholds apply for both entering and exiting.

Lines 457-473 (continuous vs. abrupt experiment methods): This should be its own subsection. Was fire (another "abrupt" source of mortality) disabled for this experiment?
**Response** : in this study fire is not use as abrupt mortality

**Corrected:** Line 474: Shouldn't this be "Qualitative"?

**Corrected:** Line 482: Table 6 should be Table 5.

**Corrected:** Line 487: "heterotrophic respiration, and disturbance."

Lines 488-489: What about emissions from combusted biomass?
**Response:** fire is not included in our simulation.

**Corrected:** Line 492: "windthrown" should be "windthrow"

Fig. 3: Table 5 describes results, not the criteria for delineating stages. What "lei panel" is being referred to? "i.e." should be "e.g." "In the lei panel… outbreak stages." These sentences can be simplified for clarity by saying that the rows are sites and the columns are windthrow intensities. Is there any significance to "small," "medium," and "large"? How are they defined? Y-axis label: Delete "gradient". What is the significance of the one plot that is circled with an arrow and labeled with "1" and "12 years cycle"? I think it means "Areas represent a fraction of 12-year simulation spent in each outbreak stage." This would be much clearer written out in the caption rather than hinted at on the figure. What is the significance of the dashed lines on each plot? Rightmost column label should be ">60%"•

**Corrected:** Line 495: "Back beetle" typo

**Fig. 4 Removed::** Consider labeling sites where outbreaks occurred with an icon of some kind in the subplot title. SOR: Put a vertical line where the second windthrow event occurred. Also, what intensity was that event?

**Section 2.4 has been updated :** Lines 569-571: This text should be included in Sect. 2.4.

**Corrected:** Line 519: "Back beetle" typo

**Corrected:** Line 521: Refer to Fig. 3 at the end of this first sentence to tell the reader where they should be looking.

**Corrected:** Fig. 5: X-axis label: "Cumulative" typo For consistency with other figures, please replace wind speed values in legend with wood loss values. Note that SOR had an extra windthrow event. "undisturbed" should be "less disturbed".

Table 5: This should probably be body text instead of a table. I suggest sub-headings corresponding to each row in the table, with paragraphs for each idea. (This will also make it possible to refer to line numbers in future review.) E.g., for Stage A, you'd have a paragraph for "post-windthrow temperature affects beetle dynamics" that covers the expected pattern and the results, then another paragraph for "intermediate windthrow sees the largest outbreaks." There is a fair amount of text here that doesn't fit in Results because it's purely model description (Methods), although some would fit well in Discussion. Alternatively, you could add some data from the results illustrating each of those. But without data, they don't fit in Results. Stage A:Literature: Add text about what sort of pattern is expected in terms of beetle dynamics after windthrow events of various intensities. Is it the "outbreaks most likely after intermediate events" that you

see in the simulations? Please provide a figure in the Supplement with a time series of mean monthly temperature (or some other indicator of temperature) to support assertions about post-windthrow temperatures affecting beetle dynamics. Add a note that the assertions are supported by comparing COL to SOR (COL is colder but has outbreaks at various windthrow intensities where SOR doesn't) and THA to WET (THA is warmer but has no outbreak at 12% where WET does). Stage B, Literature: "I. typographus" should be italicized. Stage C: First and last rows don't belong in the Results, as they're purely model description.Row 2, ORCHIDEE: What data support the second sentence? Missing period at end. Stages 4-6, row 1, ORCHIDEE: "extended" compared to what? It's shorter than the 25-year number from the literature. Why mention the entire range of modeled recovery when you're comparing to a specific beetle kill observation (52%)? How did Pfeiffer et al. (2011) define "recovery"? "back beetle" typo

**Response:** In response, we have decided to remove Table 5 and instead integrate its key points into Section 5.1, where we now discuss our findings in the context of Ips typographus-specific studies. This change allows for a more focused and relevant comparison, aligning with your suggestion to rely on observational studies and specific metrics relevant to Ips typographus. We've ensured that the references now more accurately support our statements, moving away from conceptual papers and reviews to more empirical and species-specific literature.

**Corrected:** Line 572: Table 6 should be Table 5.

**Corrected:** Line 586: Reference to Fig. 4 (Results) should be changed to Table 3 (Methods).

**Corrected:** Line 587: "bark beetle resistance"

Line 590: Define "elasticity"
**Response:** In line 772-773 - In this study we follow the definitions of Grimm and Wissel, 1997 for resistance, resilience, elasticity, and persistence.

**Corrected:** Line 592: First comma should be a period.

**Removed:** Lines 592-593: How does modeled "persistence" compare to the literature?

**Corrected:** Lines 598-599: "remains consistent" contradicts the differences seen at 20 and 50 years. Instead, I think you mean to say something like, "cumulative net biome production is similar after about 100 years."

**Corrected:**  Line 600: "convergence" should be "converge".

**Corrected:** Line 622: "degrees" should be "degree-days".

**Corrected:** Line 623: This citation should be converted to GMD style.

Line 633: "Outlook" should be combined into Sect. 4.4.
**Response:** we like having a separate section for future development.

**Removed:** Table S1: Please compose this page in landscape orientation to make room for more equations to be on one line. Also note that cell at row 2 column 3 has text cut off.Various subscripts are unexplained and various symbols are missing from Table 1.

**Minor Comments :**
**Comment:**  Is "total woody biomass" just wood in living plants, or does it also include (some?all?) Deadwood?
**Response:**  Clarified to exclude deadwood, with additional text revisions for transparency.

**Comment:** Lines 294-301: This explanation should come before "ORCHIDEE formalizes this dependency" at line 282?
**Response:** Adjusted the sequence of paragraphs to ensure a logical flow of information, particularly in the methods section.

**Comment:** Line 319: How is RDI calculated?
**Response:** Included the equations and a detailed explanation for the calculation of the Relative Density Index (RDI).

**Comment**: Lines 376-377: Referring to these as "bark beetle outbreak development stages" would avoid confusion with "development" in the sense of physiological growth from larva to adult, as well improve consistency with Table 2.
**Response:** Ensured consistent use of terms related to bark beetle outbreak stages, enhancing clarity and avoiding confusion with physiological development stages.

**Comment:** Lines 378-379: Does this refer to the hysteresis described in lines 264-278? If so, refer to Sect. 2.2.2 here. If not, please clarify.
**Response:** Revised the text to better convey the continuous process simulation of bark beetle dynamics, avoiding the misunderstanding that the model differs for different outbreak stages.

**Comment:** Line 407: *Act* is confusing here, because it's referring to activity both in the current year (previously in the text) and the next year (second part of this sentence). I suggest just removing it, or at least deleting the subscript.
**Response:** Removed potentially confusing subscripts and clarified the usage of the activity index in the model.

**Comment**: Line 499: Fig. 3 shows that these sites never left the endemic stage, but it's my understanding that trees can still be killed by bark beetles during that stage. If my

understanding is correct, please change "remained unaffected by bark beetles" to "never left the endemic stage" (unless you have other data, not shown, indicating that biomass loss to beetles was actually zero). If my understanding is incorrect, Sect. 2.2.5 should be improved.

**Response:** Improved the description to accurately reflect the impact of beetles during the endemic stage, based on the simulation results.

**Comment:** Fig. 6: How can the "background mortality only" treatment (i.e., no windthrow) be "after the windthrow event"?

**Response:** The term "background mortality" was replaced with "continuous mortality." This change is likely aimed at better reflecting the nature of the model conditions, where mortality is considered a constant factor. The mention of Figure 8 (updated from the original Figure 6) focusing on the contribution of outbreaks to carbon dynamics implies that the figures and possibly associated text have been revised to make the role of outbreaks in the model clearer, separate from the effects of windthrow events. Finally, we rewrite section 3.6 aimed at clarifying how the model accounts for mortality and its impact on carbon dynamics.

**Comment**: Sect. 4.2: Lines 567-571 should be moved to Sect. 2.4, because it provides great justification for what initially seemed a questionable choice. Then the remaining text doesn't really warrant a Discussion section; it's more Conclusion.

**Response:** Moved pertinent sentences to appropriate sections to enhance coherence and logical flow, particularly in relation to the justification for evaluation methods.

**Comment:** Lines 613-614: How does the study show anything about the importance of initial conditions? Where was that tested?

**Response:** We agree and We rewrite it to present this notion as an ancillary concept, albeit not substantiated by our research.

**Comment:** Conclusion: Also mention plans for more quantitative comparison against observed bark
beetle events.

**Response:** included a statement in the conclusion highlighting future plans for a more quantitative comparison against observed bark beetle events, underscoring the ongoing development of the model.

We hope these revisions address the concerns and suggestions raised by Referee #2, as we believe that they significantly enhanced the manuscript's clarity, coherence, and scientific rigor.

Sincerely,

Guillaume Marie

---

## Author Response (AR3)

Dear referee,

We would like to extend our sincere thanks to the referee for carefully reading and reviewing our manuscript. Your insightful comments and valuable suggestions have greatly improved the quality of our work. We deeply appreciate the time and effort you have put into providing such detailed feedback. We have incorporated your recommendations to enhance the clarity and robustness of our study. Thank you for your dedication and expertise

**Referee 1:**

**Please include Ips typographus in the title and abstract (see my comments on the first version about lack of applicability to other bark beetle systems).**

*Ips typographus* has been added to the title and the introduction. Also the other section where screened for a better use of the specific species, i.e., *Ips typographus,* and the more general wording "bark beetles".

**The abstract is still too suggestive of model accuracy, which was not evaluated (comparison of specific situations against observations). L 37-38 isn't very helpful; I suspect "any" model could produce a wide range of observed dynamics, depending on what parameters were used. Similar to "accurately mirroring a wide array of observed…" I think it's fair to say that by using different sets of parameters, model results spanned the range of observations, and that the model results indicated that including beetle outbreaks had a major effect on carbon dynamics**.

The abstract has been adjusted to address this comment. The sentences on model evaluations now read :

"*Simulation experiments demonstrated the capability of ORCHIDEE to simulate a variety of post-disturbance forest dynamics observed in empirical studies. Through an array of simulation experiments across various climatic conditions and disturbance intensities, the model was tested for its sensitivity to climate, wind disturbance, and selected parameter values. The results of these tests indicated that with a single set of parameters, the ORCHIDEE outputs spanned the range of observed dynamics*."

**L 43: Can the profound insights be described briefly here?**

I changed the sentence to explain why it was the main insight. I removed "profound" because it was too strong.

*"Notably, the study revealed that modeling abrupt mortality events, as opposed to a continuous mortality framework, provides new insights into the short-term carbon sequestration potential of forests under disturbance regimes by showing that the continuous mortality framework tends to*

*overestimate the carbon sink capacity of forests in the 20 to 50 year range in ecosystems under high disturbance pressure, compared to scenarios with abrupt mortality event"*

**L 117: Which is used here, coupled or uncoupled?**

I added a sentence that clearly says we use the uncoupled version. The text now reads:

*"Unlike the coupled setup, which needs to run on the global scale, the stand-alone configuration can cover any area ranging from a single grid point to the global domain. In this study ORCHIDEE was run as a stand-alone land surface model."*

**At the beginning of Section 2.4, please add a paragraph summarizing which mechanisms are included and which are not. For instance, I don't see beetle survival in winter (as indicated on L 494) (also, it appears that ibeetlessurvival in Table 1 is different from the description in Section 2.7, which appears to calculate ibeetlessurvival as a function of host availability, not winter temperature…or, if I don't understand, please add more description about how winter T is included).**

Adding such a paragraph would be redundant with section 2.4, 2.5, 2.6 and 2.7 in which the mechanisms are already described. Concerning $i_{\text{beetles survival}}$ a dedicated section 2.7 explains our approach. We now explain why winter temperature is not explicitly used in eq. 11. The relevant text now reads:

*"The capability of the bark beetles to survive the winter in between two breeding seasons is critical in simulating epidemic outbreaks. During regular winters, winter mortality for bark beetles is around 40% for the adults and 100% for the juveniles (Jönsson et al. 2012). In our scheme, this mortality rate is implicitly accounted for in the calculation of the bark beetle survival index (ibeetles survival). A lack of data linking bark beetle survival to anomalous winter temperatures, justifies the implicit approach and prevented including this information as a modulator of ibeetles survival. The latter explains why winter temperatures do not appear in eq. 11. Instead the model simulates the survival as a function of the abundance of suitable tree hosts which decreases the competition for shelter and food "*

**Providing one figure per variable that shows the different assumptions of the logistic curves (as well as the input variables?) would be helpful for readers.**

It was our initial thought to but given the numbers of variable x assumption, we choose to show only variables that are relevant to understand the effect of the assumption on the behavior of the model. Figures in S2 have been added to the manuscript that show the parameter sensitivity effect on the tree stand densities, relative density index and total above ground biomass.

**Section 3.3: Italicize variable names to be consistent?**

Done. We agree it enhances readability.

**L 570: Is the figure reference correct?**

Thanks for noticing. The reference was not correct and has been corrected to "figure 4".

**L 617: Please add panel labels (e.g., "(a)") to figures and refer to them in the main text.**

Done. We agree it enhances readability.

**L 620: Please add text noting that there is a difference between the duration and severity of bark beetle-caused tree mortality at the landscape scale versus at the stand/pixel scale. The landscape scale may have a longer duration and lower severity than the stand scale because beetles disperse across a landscape and cause mortality at different times. Such a distinction will be important in Table 3 and here in the interpretation of the model results.**

Following the comment by the referee, we understood that the initial text in section 2.5 was not clear enough and we used the wording of the referee to clarify this issue. The new text now reads:

*"At the landscape scale, which can cover areas up to 2500 km², the duration of mortality may be longer and the severity lower because beetles disperse across the landscape and cause mortality at different times. This distinction is important for interpreting model results, particularly when considering parameters like ird limit in the ORCHIDEE model. ird limit describes the proportion of trees surviving after an outbreak and should therefore be adjusted for the spatial scale of a gridcell in ORCHIDEE. In model set-up where a gridcell represents a single stand (~1 ha), ird limit should be close to 0, indicating that nearly all trees may be killed. However, in a simulation with grid cells representing 2500 km², not all trees will be killed, which is reflected in setting ird limit to 0.4"*

**L 622: What is "favorable"? For what objectives? If the authors mean comparison with Table 3, then I'm not sure I'm convinced: outbreaks can exhibit a range of characteristics. The authors may be able to eliminate some combinations of parameters as too extreme, but I'm not sure how parameters can be identified as most favorable.**

The sentence was reworded as "By comparing the outcomes of the sensitivity tests (section 4.1) to a compilation of observations (Table 3), a first estimate for several parameters was proposed (Table 4)."

**I'm less convinced of the selection of parameters in Table 4 based on comparison with Table 3. Table 3 should focus on Ips typographus, not other bark beetle species, and allow for a range in outcomes. Adding columns to Table 3 that describe 1) the parameter set yielding the best comparison, and values, and 2) the range of outcomes for the other parameter sets may help the reader understand the selection process better.**

We agree that the link between table 3 and table 4 is weak and unclear. In order the make the method clearer and reinforce the link between the outbreak criterias describe in table 3 and the parameter set chosen in table 4, we made a series of adjustment in the manuscript:

- We add a new column in table 3 call "reference range" which give a credible range use in the calculation of the 5 score (one for each criteria)
- We add a paragraph explaining how the Credible score was calculated :

*"Based on Table S1 and the reference range in Table 3, a score is calculated for each parameter set. The Credibility Score (CS) is the sum of four scores, indicating that the result falls within the four reference ranges described above and no outbreak is triggered when DRwindthrow = 0.1%. The CS is computed as follows: CS = (score1 + score2 + score3 + score4) * score5. Only parameter sets achieving a CS of 4 will be selected. If multiple values are possible for a given parameter, the most frequently selected value will be preferred."*

- We add a supplementary table (S1)  which gives all the parameter sets tested and their respective credibility score according to table 4

- In table 3, we now stress the fact that our data come from different beetle species, different host species and different climate zones in the caption of table 3, i.e., "". We agree this is suboptimal but it currently is the only approach to compile some data. Despite its shortcomings, lumping data from different species and climate regions is, however, common in literature reviews.

We hope that with these changes. Our choice seems less subject to interpretation while referring clearly to a protocole.

**Figure 1. "host susceptibility" instead of "host weakness" will increase reading comprehension. Same with text in the main body.**

Done.

**Why does host weakness decline? Isn't this an external factor associated with climate and age/density? Does this result from killed hosts that relieve density competition?**

Yes, when the density is declining from the outbreak itself the competition between trees for light and nutrients declines too. As a consequence the host weakness decreases. In our model it is the main reason why an outbreak stops. This explanation has now been added to the caption of Fig 1.

*"Figure 1 : Dynamic interplay of the different host and beetle characteristics  during a bark beetle (Ips typographus) outbreak. The time window spans four outbreak development stages: build-up, epidemic, post-epidemic, and endemic. The curves represent key characteristics,*

*showing the growth in beetle population  and subsequent decline in host population. Ihosts dead characterizes the presence of defenseless uprooted or cut spruce trees; ihosts alive, characterizes living spruce trees that could become hosts for the bark beetles; ihosts susceptibility, susceptibility of spruce trees to bark beetle attack; ibeetles mass attack, quantifies the capability of the bark beetles to mass attack; ibeetles survival, characterizes the survival of bark beetles. Host and bark beetle characteristics are detailed in the subsequent text.  When the density of the host trees is declining due to an increased host mortality from the bark beetle outbreak itself, the competition between trees for light and nutrients declines as well. As a consequence, the host susceptibility decreases which in ORCHIDEE is the main pathway for an outbreak to move back to the endemic phase. After 1 year the wood from a storm is not fresh enough for bark beetles to breed in. In ORCHIDEE, the bark beetle population needs to be capable of mass attacking living trees within a year to make the transition from the build-up to the epidemic phase. "*

**Why does hosts dead decrease during build-up and appear to be zero during the outbreak**?

We only use the woody biomass of hosts that are dead during the current years by other sources of mortality than bark beetles. After 1 year the wood from the storm is not fresh enough to host bark beetles which tend to favor freshly dead trees.

This is now explained in 2.7 at line 399-400 "However, after a year, this substrate becomes unsuitable for breeding and is excluded from the ihosts dead calculation." and has been added to the caption of Fig. 1 (see previous comment for the exact wording)

**Please define the variables in the caption.**

Done

**Figure 2. Having a flow diagram is a good idea. This diagram is very hard for me to understand. The reader cannot discern the meaning of the variable names. Please add them to the caption. Add boxes/ovals/circles that indicate (separately) actions/state variables/etc.**

Figure 2 has been improved.

**What is the difference between the red and black lines?**

Black lines denote intermediate variables such as indices. For the red lines the source is always a variable that describes forest structure such as RDI. The flow diagram was redesigned to better distinguish between variables internal to the ips typographus model and variables that come from pre-existing ORCHIDEE functionality.

**It would be helpful to have the processes (mechanisms) grouped together with a surrounding labeled box.**

The flow diagram was redesigned and mechanisms have been grouped (shown in the same box) to strengthen the link with the text.

**Table 3: Please use a different word than "measure" for model output, which is not a measurement.**

Done

**Also, because this study is no longer an evaluation against measurements, please use a different term than "how to check", such as "how to calculate".**

Done

**Figure 3. I assume that DRwindthrow is a parameter specified before a model run…please be explicit in the caption. Is this a relationship from an equation or an emergent property that results from a model run? Also, it would be helpful to readers to provide brief explanations of that parameter and ihostsdead in the caption.**

As now explained in section 3.5 line 560-562 windthrow damage has been forced in order to focus our study on bark beetles. A paper in which storm damage is an emergent property that results from a model run in under preparation and will be submitted in the coming months. The text now reads "In this simulation experiment, the amount of fresh dead tree hosts (Nwood) used by the bark beetles to breed was controlled by modifying the maximum damage rate of a windthrow event (DRwindthrow) in ORCHIDEE. Seven DRwindthrow were simulated (i.e, 0.1%, 5%, 7.5%, 10%, 15%, 20%, 35%)."

A short explanation has been added to the caption of Figs X. and Y to better explain the meaning and values of these parameters.

**Figure 8: What are the asterisks?**

I add the Wilcoxon test explanation into the caption.

**Also, please provide a brief explanation of the three versions (or frameworks? pick one term).**

Done

**Adding panel labels (e.g., "(a)") to all figures and referencing them in the captions and main text will minimize reader confusion.**

Done

**There are multiple typos and writing and citation style errors in the manuscript.**

We used this revision to carefully check the spelling, grammar and terminology used in the manuscript. The only native English speaker among the authors has left research which means that we have to rely on software tools to check grammar.

**Referee 2**

**Substantive comments**
**• Figs. 1 and 2 are very low-resolution and text is hard to read. Fig. 3 also has weird text but it's larger so not as much of a problem.**

New higher quality figures were prepared.

**• Line 265: How is this limited to between 0.5 and 1? Looking at the subsequent equations I don't understand.**

Thanks for noticing. This sentence was right in the previous version but given the equation evolved any value from 0 to 1 is possible. I change the sentence on line 265 to reflect this change.

**• Line 281 (Eq. 5a): The "1 –" in the exponential seems to make this function point the incorrect way, with $i$ lower (i.e., stand is less attractive to beetles) when $PWS$*+, is high (i.e., a strong drought happened recently) and vice versa.**

In orchidee PWS=1 means no stress, this is the reason why with a 1- but I understand your concern. To make it clear and to be consistent with the code, I decided to add a sentence explaining this around line 281.

**• Line 285 (Eq. 5b):**
**-What is "nb age class"?**
**-Why is age class 1 the maximum considered?**
**-(The above two questions also apply to Eq. 6b.)**

I clarified this issue at line 285. The text now reads:

$$"PWS_{max} = \sum_{nb\,age\,class=3}^{age\,class=1} max(PWS_{spruce,\,n}, PWS_{spruce,\,n-1}, PWS_{spruce,\,n-2}) \times \frac{F_{spruce\,class}}{F_{spruce}} \qquad (5b)"$$

$$``i_{rd\,spruce} = \sum_{nb\,age\,class=3}^{age\,class=1} \frac{D_{age\,class}}{D_{max}} \times \frac{F_{age\,class}}{F_{spruce}} \qquad (6b)"$$

**-Are the *PWS* values here averages? Over what time periods?**
**-If *PWS*#-./0' is this year's plant water stress, *PWS*#-./0',)23 is last year's, etc., then**
***PWS*#-./0',)24 being included means that it's actually looking over the past four years, not three as mentioned at line 279.**

We correct for PWS$_{spruce,n-2}$ because ORCHIDEE only checks 3 years. We also add the description of PWS$_{spruce}$ at line 289.

**• Lines 358-361: This text suggests that Fig. 1 shows both a return to endemic stage and an evolution to epidemic, but it looks like it only shows the latter.**

The referee is correct. I removed the reference figure 1.

**• Line 393: Is *B*wood live biomass only?**

Yes, it is used as a reference in order to reflect that when Nwood = Bwood the susceptibility reaches its maximum. That is arbitrary but we think it is reasonable to think that above 50% of the living trees kill, there is too much substrate than an endemic bark beetle population can eat in 1 year.

**• Lines 451-452: Why is this index calculated separately for each age class whereas the indices contributed to by Eqs. 5b and 6b are calculated as cross-age-class averages?**

In order to consider different mortality rates between dominant, subdominant and understorey trees (here represented by the 3 age classes) the indices are calculated for each age class. In equations 5b and 6b indices are calculated across age-classes to account for stand susceptibility. Eq 5b for exemple is used as a proxy of stand healthiness for beetle attraction.

**• Lines 490-491: What does "acclimation" refer to in this sentence?**

It is now clarified that "acclimation" means "change their behavior or their resistance to external stressor such as winter temperature, of the bark beetle population to each location."

**• Line 517 and in Results: Ranges don't make sense in this context. What exact values were tested? Line 516 says that only three values were chosen.**

We reformulate that paragraph in order to clean the method. it is now read :

*"For each of the four variables, three distinct values were assigned to two parameters labeled "Shape" and "Limit". The Shape parameter determines the shape of the logistic relationship, with three values tested: (a) Shape=-1, yielding a linear relationship, (b) -5<Shape<-30, resulting in a logistic curve, and (c) Shape=-500, turning the logistic relationship into a step function. For the logistic curve, the exact Shape value between -30 and -5 is chosen according to each index under study:   (1) Ssusceptibility = -5; (2) Sactivity= -5; (3) Smass attack= -30; and (4) Sgeneration=-5 . For Smass attack, a higher value has been chosen because the slope of the logistic curve has a significant impact in order to trigger an outbreak."*

**• Lines 559-560: Is this 20% number referring to $max65""\&$ ?**

Yes, I have clarified in the text.

**• Lines 566-571: How do you calculate the number of generations per year?**

It is the exact same equation as in the original paper from Temperli et al. 2013. The equation is now given on line:

$$DD_{eff} = \sum_{i=1}^{n_{diapause}} (T_{opt} - T_{min}) * e^{(0.0288* T_{bark,i})} - e^{(0.0288* b_{eff} - (40.99 - T_{bark,i})/3.59)} - 1.25 \quad (10)$$

Where $i$ is a day, $n_{diapause}$ is the number of days between the 1st of january and the day of the diapause. $T_{opt}$ (30.3°C) is the optimal bark temperature for beetles development and $T_{min}$ (8.3°C) is the temperature below which the beetles developpement stop. $T_{bark, i}$ is the average daily bark temperature. $T_{bark, i}$ is calculated as the daily average air temperature minus 2°C. All parameters values are taken from Temperli et al. 2013

**• Lines 573-581 and subsequently: "Control" and "climate experiment" feels weird. For the sites, it's more of a climate gradient that starts with HYY and continues through HES/FON. There's nothing a priori different about HYY that makes it a "control" and the other sites "experimental." Similarly for the windthrow damage rates: Maybe if the lowest value was 0 that would be a control, but it's not. (Note that I'm not saying true control treatments are needed—I don't!)**

I understand your concern, HYY should be considered as a reference location for no outbreak and 0.1% damage has a small enough damage rate that should never trigger an outbreak. This comment was addressed by changing the wording throughout the text.

**• Lines 588-593: Note that (1) is the default or previous ORCHIDEE setup.**

Yes, I make it clearer

**Sect. 4.1: I'm unclear on how the sensitivity tests indicate anything about "false positives" due to the calculation of any specific parameter. Weren't all experiments done with all parameters?**

False positives are directly deducted from the control (now renamed as reference for no outbreak). If an outbreak is observed for HYY or 0.1% damage rate we can consider it as false positive but not in its statistical meaning but more as a boundary that ORCHIDEE should not cross. So given the confusion surrounding the term "false positives" in this context, the wording was replaced by "improbable outbreaks" throughout the manuscript.

**Table 4: Missing $S0"*-'$7$7")$ .**

Done

**Lines 683-684: This sentence is confusing. "Turning back" from what? If the "tipping point" is 9 kgC/m2, what is the "threshold"?**

Line 690-694: I have rewritten this part because some of the statement was wrong; It is now more clear since 9.0 is not considered as the tipping point that led to an epidemic.

**Lines 684-685: REN clearly reaches some "tipping point"—i.e., it reaches a minimum and then recovers. What's so special about 9 kg/m2?**

At line 691 : "at a biomass level around 9.000 gC.m-2 equivalent in our simulation to an $i_{rd\ spruce}$ = $i_{rd\ limit}$=0.4, there's no turning back until that threshold is passed."

**Fig. 7: Add shading or some other indicator of where the outbreak (as defined according to Table 3) begins and ends**.

We added a grey area that represents the epidemic phase in each panel/site.

**Lines 721-722: "a more resilient recovery" doesn't seem right for the continuous case, where there's no specific event to "recover" from**.

I agree that by rereading the sentence I realize that a mistake was made because it makes sense for the "abrupt" configuration but not for the "continuous" one. I decided to remove this sentence.

**Lines 752-753: Unclear how this sentence is related to the rest of the paragraph (snags)**.

I have removed this sentence because none of our figures suggest the mechanism even if it is indeed happening.

**Line 767-775: Terms should be defined in this paper. It's not really helpful to cite a paper for readers to look up the definitions, especially after the terms are used. Ideally definitions should be mentioned in the introduction or methods so that they're defined before they're used—note that "resilience" is used in the results section**.

The definitions were added in the text.

**Lines 777-799 (Sect. 5.3): Discussion (and also Conclusion) should mention that this depends on spatial scale and model use. If one is looking at global-scale NBP, there might not be as much of an impact of including abrupt disturbances because they're happening constantly somewhere**.

In this study, we only check the temporal aspect of the abrupt versus continuous mortality. Increasing the spatial domain is one of the main objectives of the second paper.  This is the reason why we did not cover this aspect in this study.

**Technical corrections, etc.**
• Line 257: "analogue the the" should be "analogous to the". done
• Line 279: "average" is sort of confusing here. It looks like it's the (weighted) average across different age classes of the maximum water stress in each age class. Maybe just delete "the average" from this sentence to make it work. Changed to maximum
• Line 288: A paragraph break before "In addition to drought" would help break this section up a bit. Done
• Line 308: "mean quadratic" should be "quadratic mean". done
• Line 323 (Eq. 7b): "none" should be "non". done
• Line 372: Is "excess" the right word? replace by stimulation
• Line 409: Incomplete sentence: "The amount of suitable tree hosts." done
• Line 464: "bettles" typo in subscript. done
• Line 501 (Table 2):
o I thought the spelling looked weird for "Wetstein," and indeed, it doesn't
actually seem to be a place in Germany. The listed coordinates point to
Třeboň, Czechia, about 400 km southeast of the similarly-named German
mountain Wetzstein. The listed coordinates do, however, look similar to
those of the CZ-Wet (CZECHWET) site near Třeboň. done
o I can't find the HES site in the FLUXNET site table, or any site with similar
coordinates. It's possible it's just missing from that table, but given the above issue it warrants a
double-check. HES was removed from FLUXNET in 2015 but we still use the forcing from
1997-2014.
o The FON latitude appears incorrect according to the FLUXNET site table. done

o REN site latitude should round up to 46.6°N. (Check rounding of other sites as well.) done
Line 570: Reference to Fig. 3 should be to Fig. 4. done

Figs. 3 and 4: These figures refer to "Control" and "Climate experiment" which isn't explained until later in the text. Done

Line 606: Delete ")". done

Line 607: Delete ")". done

Line 617 and elsewhere: Instead of having to say "4th row, 2nd column," add subplot labels a-h. done

Lines 678, 683: Thousands separator should be comma, not period. So 9,000 instead of 9.000. Or to avoid cross-cultural confusion, just use 9 kg instead of 9,000 g. Done

Line 718: "Abrupts" typo. done

Line 731: "maining"? replace by impact

Line 761: First comma should be a semicolon. done

Line 767:

o "resistance" should be defined. It's not really helpful to cite a paper for readers to look up the definitions, especially after the term is used. done

o "locations" should be "location" (or just deleted) ??

Line 810: "Ips Typographus" should be "Ips typographus" done

Lines 811-812: Latin name should be italicized done

Line 860: Same DOI repeated twice. Maybe these both will be replaced with the new DOI the authors mentioned in their cover letter? data and script have been pulled together for an unique DOI

---

## Author Response (AR4)

Dear Referee,

Thank you very much for this third round of comments. I have carefully addressed all your feedback,
and I hope that the manuscript is now progressing toward its final form.

Fig. 1 caption: "After 1 year the wood from a storm is not fresh enough for bark beetles to breed in, so *ihosts_dead* goes to zero." added

Table 1:
o Alphabetize added
o Inconsistent variable capitalization (here and in text) solved
o Inconsistent use of underscores vs. spaces in subscripts (here and in text) solved
o Add "susceptibility" to description solved

L273: "the max" should be "the maximum". solved

L277–281 (Eq. 5b):
o I'm afraid the added information about age classes just served to confuse me further. I think it should be reworked like this? solved
where $a$ is age class and $y$ is the current year. (Similar changes should be made for Eq. 6b.) added

o L280: Also mention that PWS is 1 when plants aren't stressed at all. added

L452: "index of weakened trees index"—delete second "index" Fig. 2 solved

o Caption: Refer the reader to Table 1 for variable name definitions. added

o Rounded rectangle whose left side comes between, e.g., PWS_max and the "5" arrow should be deleted. It breaks the connection between the variables on the left and the arrows on the right, and it doesn't seem to add anything useful to the figure. removed

Table 3 caption:
o "spacity" should be "sparsity" or (preferably) "scarcity". solved
o "but see table s1"—why "but"? removed

Table 3: "A severe bark"—delete "bark". removed

Sect. 3.4:
o Note that scores are either 0 or 1. added

o "is the sum of four scores" isn't quite right, since that sum is multiplied by

score5. changed to "is the sum of four scores time the fifth score"

o Discussion of "four scores" in first paragraph and bulleted list neglects score5—why? bullet point added saying "No outbreak is triggered when DRwindthrow = 0.1%. This characteristic is mandatory in order to avoid outbreak for no specific reason (score5). "

Sect. 4.2: Refer to Table S1, but note that it only contains the variables from Table 4 that were sensitivity-tested. added

L724: What is the "t" in this sentence? removed

Table 4: Please alphabetize. done

Sect. 5.3: Previously, I requested: "Discussion (and also Conclusion) should mention that this depends on spatial scale and model use. If one is looking at global-scale NBP, there might not be as much of an impact of including abrupt disturbances because they're happening constantly somewhere." The authors replied that a spatial analysis is the focus of their next paper, not this one. That's fine, but the point I raised should still be mentioned here. It's fine to say what the paper plans are as well, if the authors would like.

added at lines 836-338 : "However, it is important to note that the global-scale NBP might not be significantly impacted by abrupt disturbances compared to continuous ones. This is because the scale is so vast that abrupt mortality events are constantly occurring somewhere on the planet, which smooths out the effects illustrated in Figure 8."

Line 804: "locations" should be "location" (or just deleted) deleted

Supplementary figures and tables: S in numbering should be capitalized. (E.g., S4, not s4.) solved

Figs. S2-4 have the same caption. They seem to diaer only in the variable being plotted on the Y axis, so please add that information to the caption. Then the captions of Figs. S3 and S4 can be simplified by saying something like "As Fig. S2, but for total biomass." solved

Throughout main text and supplement:
o "access" should be "assess" solved
o "developpement" should be "development" solved

Table S1

Red/green is not colorblind-friendly. Other colors would be better, but avoiding color would be best. It's also confusing because some of the green values are also reference values, which should be black.

Here's one suggested setup that would be clearer and more accessible: Italics to indicate "reference values" (which would more clearly be referred to in the caption as "variables held constant for each sensitivity test"), underlines to indicate values of sets scoring 4, and bold to indicate the values ultimately chosen. See screenshot below. solved

Caption:
• "ips typographus" should be "Ips typographus" solved
• "The parameter set in green": "set" should be "sets".solved
• "32" should be "36" solved
• What does "selection" mean here? Replace by "the 36 parameter sets used to …"